# A framework for on-implant spike sorting based on salient feature selection

MohammadAli Shaeri [1,2] & Amir M. Sodagar [1✉]

On-implant spike sorting methods employ static feature extraction/selection techniques to minimize the hardware cost. Here we propose a novel framework for real-time spike sorting based on dynamic selection of features. We select salient features that maximize the geometric-mean of between-class distances as well as the associated homogeneity index effectively to best discriminate spikes for classification. Wave-shape classification is performed based on a multi-label window discrimination approach. An external module calculates the salient features and discrimination windows through optimizing a replica of the on-implant operation, and then configures the on-implant spike sorter for real-time online operation. Hardware implementation of the on-implant online spike sorter for 512 channels of concurrent extra-cellular neural signals is reported, with an average classification accuracy of ~88%. Compared with other similar methods, our method shows reduction in classification error by a factor of ~2, and also reduction in the required memory space by a factor of ~5.

[1] Department of EECS, Lassonde School of Engineering, York University, Toronto, ON, Canada. [2] School of Cognitive Sciences (SCS), IPM-Institute for Research in Fundamental Sciences, Tehran, Iran. ✉email: sodagar@eecs.yorku.ca

In the realization of brain implants and neural prostheses, one of the main challenges is to increase the number of recording channels. This is mainly because of the significant increase in the need for power consumption, data telemetry bandwidth, and also enlarged physical dimension of the neural recording implant. On-implant spike sorting is one of the possible steps towards overcoming such challenges by efficient data reduction.

Generally, spike sorting can be performed through the following general steps: (i) filtering the raw neural signal (from 0.3 to 6 kHz)[1] to preserve only the useful frequency content of neural spikes, (ii) detection of spike events upon the firing of neurons, (iii) extraction of spike wave-shapes from the filtered neural signal (for details of our spike detection and extraction method, refer to our previous work[2]), (iv) temporal alignment of the spike wave-shapes, to avoid additional hardware cost, in this work spike wave-shapes are aligned to the detection (first threshold-crossing) points, (v) mapping of the extracted spike wave-shapes into a feature space, known as feature extraction, this step is to enhance the discrimination between spikes and noise, and also between different spike classes (also referred to as between-class variability), (vi) selection of a minimal subset of features, known as feature selection, in order to reduce the dimensions of the data being processed, and (vii) classification or clustering of the wave-shapes into different spike classes as isolated units.

From the standpoint of computational load (and consequently hardware complexity), most of the traditional spike sorting algorithms are too heavy to be implemented on neural recording implants. To efficiently realize spike sorting on such implants, one solution is to reduce the dimension of the data being recorded. For on-implant online spike sorting, peak values and timings[3–11], and zero-crossing points[12] have been selected as simple and informative geometric features to sort spike classes. Furthermore, to enhance the discrimination between different spike classes, hardware-efficient mathematical transforms such as derivative transforms[3–5,7,10,13–15] and four-level Haar wavelet transform[8,9] have also been used for feature extraction on brain implants. To make the spike sorting procedure complete, on-implant classification of spike wave-shapes has been realized using distance-based classification[16–18] and oblique decision tree classification methods[8,9].

## Results

### System description

In this work, we propose an automated method for online spike sorting dedicated to high-density, high-speed brain implants. The proposed method needs to be simple, agile, and reconfigurable, and at the same time should be physically implemented in compliance with physical and electrical limitations of brain implants. Computational load of the existing spike sorting procedures usually entails technical challenges such as hardware complexity, power consumption, and computation speed. The technique we propose overcomes these challenges by shifting the computational load from the implant to the external side of the system where complexity of the algorithm and its hardware implementation is not as important.

Traditionally, fully implantable neural recording systems comprise an implantable module and an external module communicating with each other via a wireless link. As depicted in Fig. 1, the miniaturized implantable module records intracortical neuronal activities on multiple channels using a penetrating microelectrode array. The implantable module is in charge of the recording of neuronal activities. The external module, in general, communicates with the implantable module through bidirectional wireless communication. It receives the recorded information from the implant, stores the recorded data, performs signal

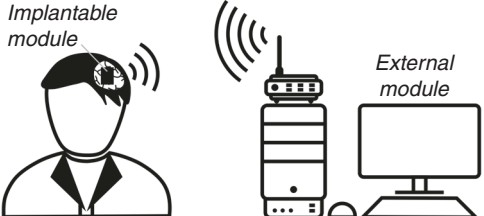

**Fig. 1 Illustration of a brain-implantable system, including an implantable module and an external module communicating with each other through wireless connection.** In general, the implantable module is configured by the external module through wireless connection. It may also wirelessly receive electric power from the outside. The implantable module comprises a microelectrode array for extracellular recording of neuronal activities, and the electronic circuitry supporting its function. The electronic part usually includes a neural signal preconditioning block, which pre-amplifies the signals sensed by the electrode array, filters out the out-of-band frequency components, and finally digitizes the signals using per-channel or time-shared analog-to-digital converters. After digitization and some signal processing tasks, the recorded signals are telemetered through a wireless link to the external module.

processing tasks, and possibly sends data/configuration information back to the implant.

### The proposed spike sorting framework

A conceptual block diagram of the system, on which the proposed method is realized is shown in Fig. 2a. In this scheme, the implantable module records neuronal activities, runs digital signal processing procedures, and finally performs online unsupervised spike sorting. The external module is in charge of the calculation of the parameters using which the on-implant online spike sorter (OSS) is configured and calibrated.

To significantly reduce computational and hardware complexity on the implant, the proposed spike sorting method is divided into two phases: an offline initial training phase implemented on the external module, and the main online spike sorting phase realized on the implant. In the real time and with area- and power-efficient hardware on the implantable module, what remains on the implant is a compact, low-power, and agile OSS, which is configured using the results of the offline training phase received from the external module. The key value of the proposed spike sorting technique is in its potential to allow for a power- and area-efficient hardware implementation that operates in the real time on a high-density neural recording implant. Prior to the start of the operation of the on-implant OSS, first, the system telemeters neuronal activities (spikes) on all the channels to the external module through wireless connection. An unsupervised offline spike clustering block (based on the silhouette statistic[19–21] and k-means clustering algorithm[21]) on the external module then labels the spikes received from implantable module. As shown in Fig. 2b, a shadow spike sorter on the external module (which includes an identical model of the on-implant OSS) receives both the spikes and the associated labels, and is optimized to perform the proposed spike sorting algorithm. The OSS model parameters are then sent to the implantable module in order to configure the on-implant OSS. After configuration, the on-implant OSS will be able to perform spike sorting on the live stream of the neural signals being recorded.

### Salient feature selection

As will be discussed later in this paper, existing spike sorting techniques commonly use specific geometric features for spike wave-shape isolation. Although such features offer the advantage of straightforward mathematical

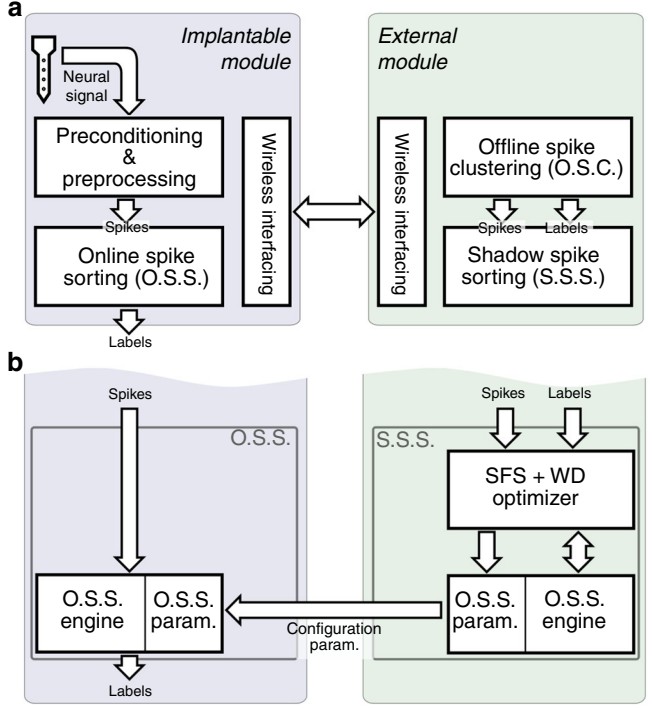

**Fig. 2 Functional diagram for the implementation of the proposed online spike sorting approach on an implantable neural recording system. a** The Implantable Module contains an online spike sorting module that needs to be configured by a shadow spike sorter block on the external module. In the training phase, the offline spike clustering block on the external side of the system receives a long-enough recording on all the channels, and performs unsupervised spike sorting in order to identify all the units (spike classes) in the brain area under recording. Outcome of this offline spike clustering is the identified spike classes and the associated labels, which are used to optimize a model of the proposed method (the shadow spike sorter). **b** The shadow spike sorter block is indeed a model of the salient feature selection method followed by window discrimination (SFS + WD), which is optimized using the spike class information it receives from the offline spike clustering block. After optimization, settings and functional parameters of the shadow spike sorter are used to configure the main online spike sorter on the implant.

formulation and rather simple hardware implementation, they do not necessarily guarantee maximal discrimination between spike classes. The proposed online spike sorting technique is based on finding a minimal set of geometric features, hereafter referred to as salient features, that maximize the discrimination between spike classes. Each and every spike class is discriminated from all other spike classes (multi-label classification[22]) using a subset of salient features in the salient feature space.

As a measure for the extent of the overall separation of the class of interest (#i) from all other classes, saliency of that class ($\varsigma_i$) is hereby defined in such a way that it expresses both its discrimination from all other classes and the extent of the homogeneity of the distribution of all other classes (with respect to class #i). To quantify the saliency of class #i, the former is measured by the geometric mean of the associated distances, and the latter is quantified as the ratio of the geometric mean to the arithmetic mean of the same distances. It should be added that according to the definition presented in Beauchemin et al.[23] and Woodhouse et al.[24], the signal space is considered homogeneous with respect to a certain class if that class is equally separated from each and every other class in the signal space. In the simplest scenario, where the feature space is one-dimensional, the

saliency of class #i ($\varsigma_i$) is, therefore, formulated as

$$\varsigma_i = \frac{\left(\prod_{j=1(j\neq i)}^{N_c} (d_{ij})^{P_j}\right)^2}{\sum_{j=1(j\neq i)}^{N_c} P_j \times d_{ij}}, \quad (1)$$

where $i$ is the index of the class of interest, $d_{ij}$ is the class #i and #j discrimination index (refer to "Methods"), $P_j$ is the relative probability of class #j, and $N_c$ is the total number of classes.

In general, the concept of class saliency can be extended to a $K$-dimensional space. For the $k$th feature ($k = 1, 2, \ldots, K$), $\sigma_i[k]$ is defined to express the saliency of class #i from all other classes (refer to "Methods"). From among all the features in the $K$-dimensional feature space, the most salient feature (MSF), $k_i^1$, is introduced here as the feature that distinguishes class #i from all other classes with the highest possible class saliency. This is, indeed, the first member of the salient feature set, determined by spanning the entire ($K$-dimensional) feature space. Index of the MSF for the class #i is, therefore, expressed as

$$k_i^1 = \underset{\kappa \in \{1,2,..K\}}{\arg\max} \{\varsigma_i[\kappa]\}, \quad (2)$$

Selected from among the remaining $K$-1 features in the feature space, the second MSF (2nd MSF), $k_i^2$, is the most uncorrelated feature to the MSF ($k_i^1$) that best isolates class #i from the rest of the signal space. The 2nd MSF is mathematically determined as

$$k_i^2 = \underset{\kappa \in \{1,2,..K\}}{\arg\max} \{\varsigma_i[\kappa] \times (1 - \rho_i(\kappa, 1))\}, \quad (3)$$

where $\rho_i(\kappa, 1)$ indicates the correlation between the $\kappa$th feature in the feature space and the 1st member of the salient feature set, i.e., the MSF ($k_i^1$). The term $1 - \rho_i(\kappa, 1)$ in Eq. (3) is to ensure that the information redundancy in the salient feature space is eliminated or at least reduced. In general, the process of selecting $L$ features of the highest saliency (i.e., forming an $L$-dimensional salient feature space for a given spike class, as formulated in "Methods") is referred to as the salient feature selection (SFS) process. In a given spike sorting problem, the value of $L$ is determined by the user as a result of a tradeoff between hardware cost and the achieved classification accuracy (CA).

Figure 3a illustrates a spike classification problem with three isolated units, mean values of which are shown using red, green, and blue solid lines ($\mu_1$, $\mu_2$, and $\mu_3$, respectively). Here, each spike sample is taken as a feature. Hence, assuming that a neural spike is expressed using $K$ samples, the main feature space consists of $K$ feature dimensions. The SFS method is now used for the generation of a salient feature set, by selecting the samples that best discriminate each spike from other spikes. The MSFs (samples) shown with circles on each spike class are indeed the ones that provide the highest saliency. Figure 3b–d present the details of the SFS process in the case of this neural spike classification problem, in which the horizontal axis is the "Feature Index ($k$)". The geometric means and homogeneities of $d_{ij}$'s associated with the three spike classes are shown, respectively in Fig. 3b, c based on which class saliencies are calculated and plotted in Fig. 3d. According to the subplots shown, saliency of a unit with respect to the others ($\varsigma_i$) peaks when the product of the geometric mean and the homogeneity associated with that unit is reasonably large.

To evaluate and validate the success of the proposed concept of saliency in order to form an efficient feature set (i.e., the salient feature set), we use the Bayes classifier to complete the spike classification process. We are to show that there is a strong correlation between the saliency of the features used for classification and the CA achieved. The scatter plot in Fig. 4a presents the saliencies (calculated based on Eq. (6) in "Methods")

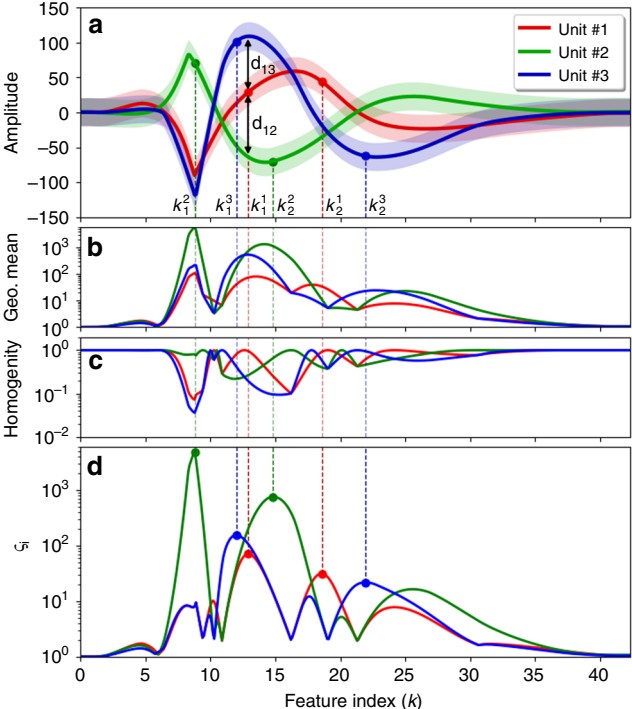

**Fig. 3 Details of salient feature selection. a** A typical spike sorting problem with three spike classes. **b** Geometric mean values for the exponential class discrimination indices $d_{ij}$'s. **c** Homogenity of $d_{ij}$'s. **d** Class saliency ($\varsigma_i$), which is indeed the product of geometric mean values of $d_{ij}$'s and the associated homogenties. The criterion for identifying the feature that best distinguishes a given spike class from all other classes (the most salient feature) is where saliency of that class is maximized. The samples (features) shown by solid circles on each spike are the most salient features, using which that spike is distinguished from other spikes in the best way.

versus the Bayes CA for the three classes shown in Fig. 3a. It can be obviously seen in this figure that the chance level of each class (illustrated by dotted lines) contributes to the associated accuracy of classification. However, this contribution can be somehow misleading when evaluating and comparing classification accuracies for classes of different chance levels. To have a fair comparison, we therefore eliminate the influence of the chance level from the CA. Hence, as a modified measure, the chance-level-independent CA ($CA_{CLI}$) is proposed as

$$CA_{CLI} = \frac{CA - Chance}{1 - Chance}, \qquad (4)$$

in which CA is the classification accuracy with its conventional definition, and Chance is the chance level associated with each class. The same comparison after the elimination of the chance level from the classification accuracies of the clusters is presented in Fig. 4b. The distribution of the data points in this plot is an indication of a meaningful statistical correlation (correlation coefficient ≈0.8) between the log-saliency of features and the $CA_{CLI}$ they result. Therefore, it can be concluded that the saliency metric proposed in this work is an efficient criterion for the selection of a subset of features for successful classification.

**Window discrimination**. For online on-implant spike sorting in this work, we use a window discrimination (WD) method in the salient feature space. The WD method benefits from efficient hardware implementation, which is of crucial importance in the design of brain-implantable microsystems. In order to classify each and every unit, one discrimination window is assigned to each class in the associated salient feature space. Specifications of the four borders of each window are determined in the offline phase, and are subsequently stored in the on-implant spike sorter for online spike classification.

Number of dimensions in the salient feature space is an important aspect of the feature selection method that needs to be decided upon. In this work, a salient feature space with two dimensions leads to satisfactory isolation of units with the

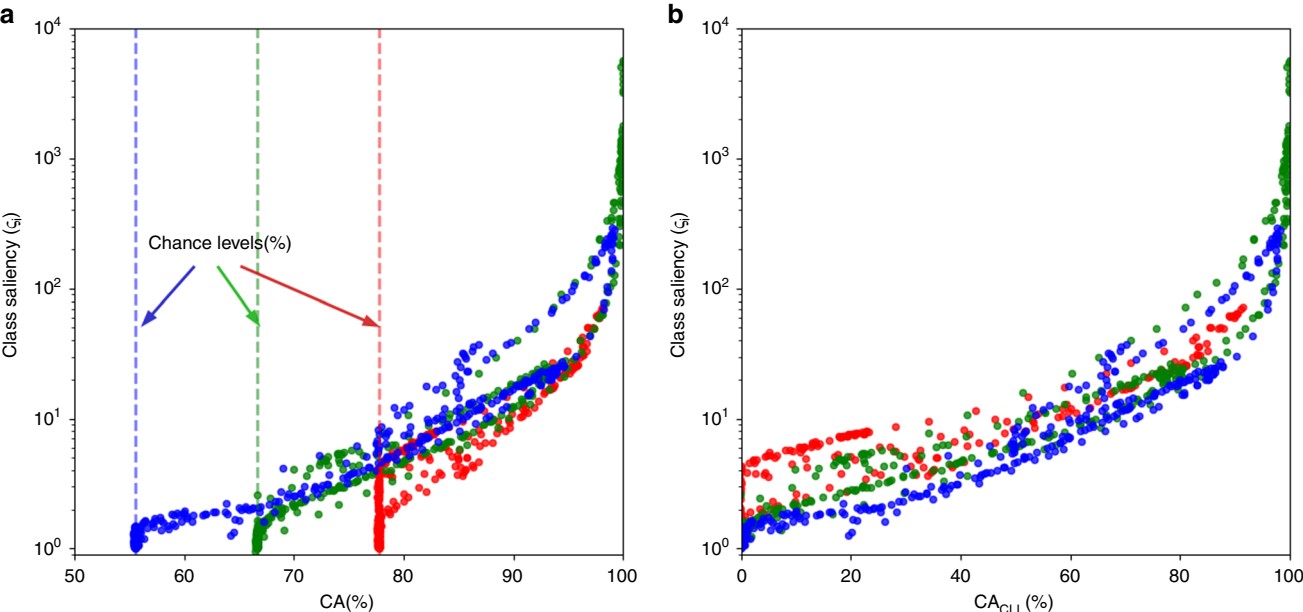

**Fig. 4 Class saliency versus the classification accuracy achieved using a Bayes classifier on the entire feature space (a) before and (b) after the removal of chance levels.** In **a**, a minimum of chance (%) is guaranteed for the CA (%) associated with each unit, which is indeed not coming from the success of the classification performed. By using $CA_{CLI}$ (%) on the horizontal axis in **b**, the chance of occurrence for the units present in the test data does not contribute to the measurement of the classification accuracy (hence taking on values in the entire range of 0–100%).

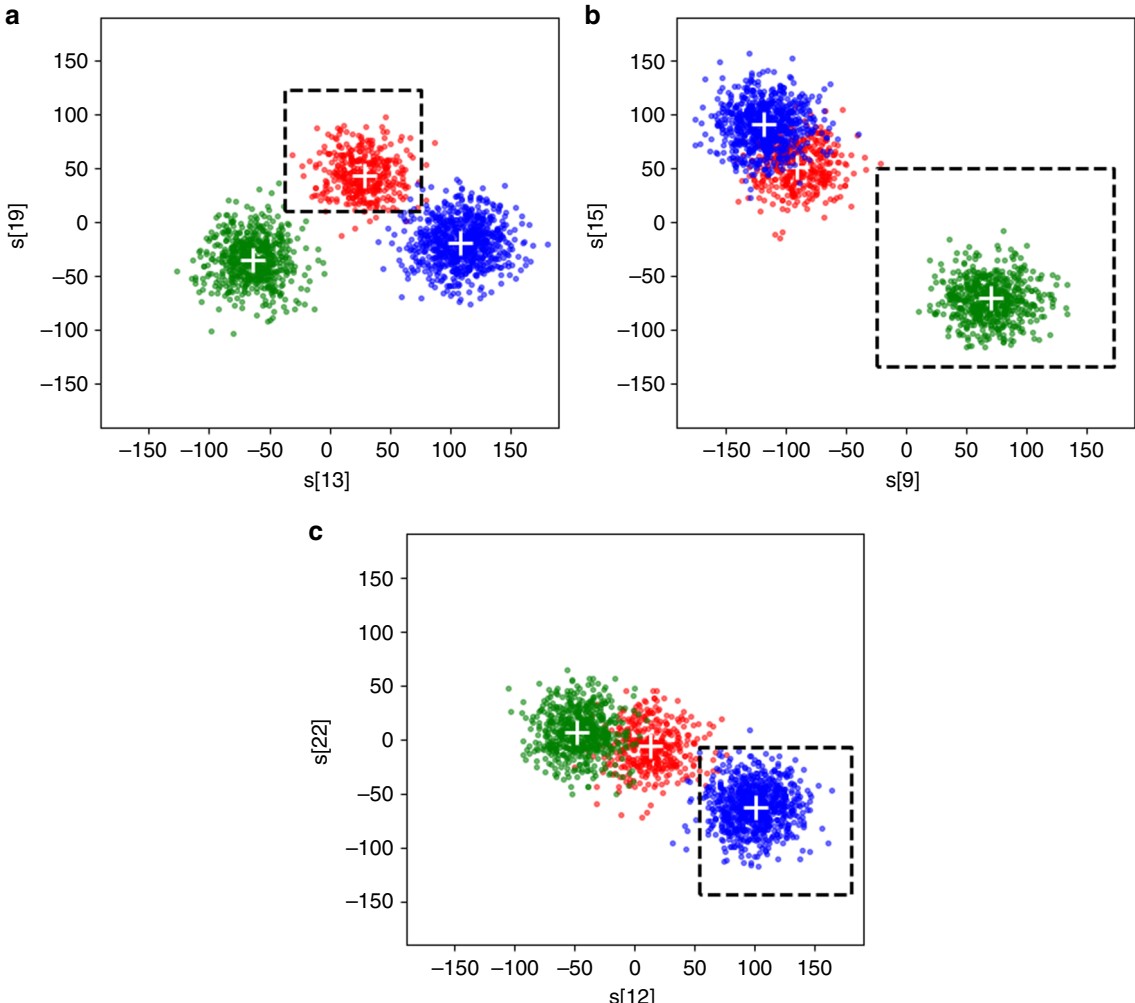

**Fig. 5 Multi-label WD classification of spike wave-shapes in their associated salient feature space for the classification problem described in Fig. 3.** According to the basics of multi-label classification, in each plot **a**–**c**, one of the clusters is isolated from the rest of the signal space. The dashed rectangles specify the discrimination windows for a pair of salient features that best confine the class of interest.

accuracy of 87.6% or higher. For the classification problem under study with three classes of unit activities shown in Fig. 3a, a two-dimensional (2D) salient feature space is defined for each class of unit activities. In this case, multi-label discrimination windows in the associated salient feature spaces are shown in the scatter plots of Fig. 5. In these plots ($s[13]$ and $s[19]$) are the salient features used for the classification of unit #1, and ($s[9]$ and $s[15]$) and ($s[12]$ and $s[22]$) are the salient features used for the classification of unit #2 and unit #3, respectively. In these scatter plots, the coordinates of the (upper bound and lower bound) for the discrimination windows associated with units #1, #2, and #3 are {(−28, 70), (−3, 103)}, {(−11, 147), (−171, 185)}, and {(54, 222), (−133, 4)}, respectively.

**Performance evaluation**. For all the tests presented in this section, the open-access data set of prerecorded spike wave-shapes[25] is used to generate the data required for both training and testing. This data set were recorded by a 10 × 10 Utah array from populations of neurons in primary visual cortex (V1) of macaque monkeys (*Macaca fascicularis*) in response to natural images. From this data set, we generated a library of ~15,000 different spike classes. Each spike class consists of hundreds of spikes with signal-to-noise ratios ranging from ~0.3 to ~22 (with the average value of ~4.5). Sampling rate of the recordings is 30 k sample/s,

with the resolution of 8 bits. All the spikes extracted for classi-fication are 48 samples long (1.6 ms). For each trial, a random selection of ~1450 spike classes (units) are chosen. From this "trial library", two, three, or four units are used to train and test each channel. For each and every channel, the spikes under each unit are used for training and testing with a breakdown of 50–50%.

To evaluate the efficacy of the idea of salient features and the SFS approach proposed, the results achieved in this work are compared with the other major feature extraction/selection methods already appeared in the literature. On-implant spike sorting methods normally use specific features with straightfor-ward mathematical descriptions to classify spike wave-shapes (referred to as static methods in this work). The static techniques used for comparison include peak-to-peak amplitude of the spike and min-max of its derivative, hereafter referred to as spike and derivative extrema (SDE)[3], first and second derivative extrema (FSDE)[4], event-driven features (EDF)[11], discrete deriva-tives and their peak values, hereafter referred to as discrete derivatives extrema (DDsE)[7], zero-crossing features (ZCF)[12], minimum delimitation (MD)[26], and the Haar-wavelet-based frequency-band separation (FBS$_{HT}$) method[8,9]. All these feature extraction/selection methods (including the proposed SFS approach) are followed by the same type of classifier for a fair comparison. For this purpose, the Gaussian Naive Bayes

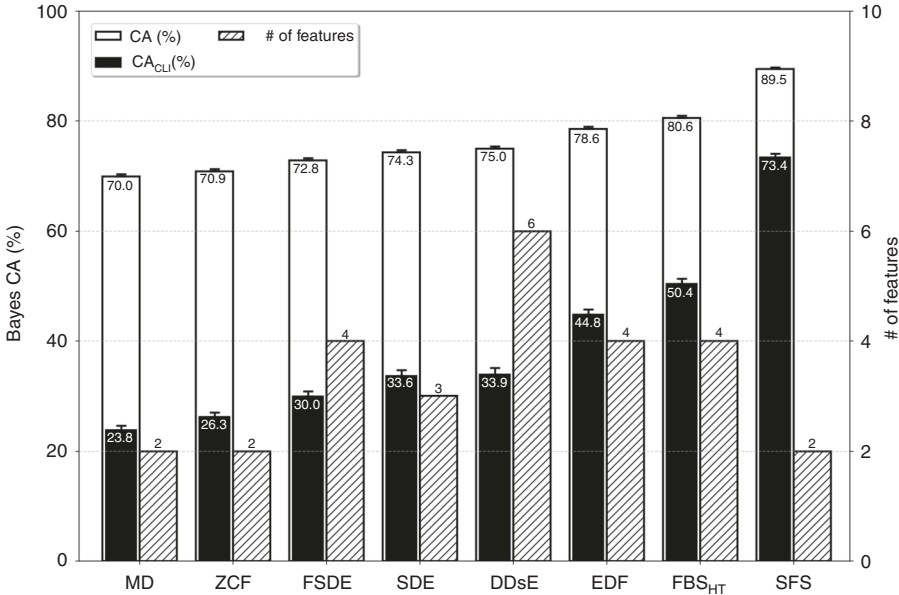

**Fig. 6 Number of features, CA, and CA_CLI for SFS and previously reported feature extraction/selection methods when followed by a Gaussian Naive Bayes classifier for all spike classes ($N = 1472$ independent spike classes including 125,120 test wave-shapes; data are presented as mean values ±SEM).** The proposed salient feature strategy (with only 2 features) results in significantly higher classification accuracy, both with and without chance level removal, compared with other existing feature extraction/selection methods (with four features in the most successful cases).

classifier[27,28] is used. This classifier only requires data statistics (mean and variance) with no need for manual setting of parameters. In this study, all the resulting spike sorters are evaluated with the same data set.

The regular CA, the chance-level-independent CA (CA_CLI), and the feature space dimension for all the aforementioned feature extraction/selection methods reported earlier in the literature of brain implants are presented in Fig. 6. Spike sorting using the SFS method proposed in this work exhibits significantly higher CA (89.5%) and CA_CLI (73.4%) than all the other methods. It is important to note that, even with as low as 2 dimensions for the feature space, the SFS-based spike sorter outperforms all other sorters from the standpoint of the achieved CA. This is translated to much less computational cost, which will lead to a significantly more power-/area-efficient hardware when it comes to on-implant physical implementation.

To evaluate the efficacy of the proposed feature selection method in introducing a more appropriate subspace of features (i.e., the salient feature space), CA of SFS followed by a Bayes classifier is compared with that of Bayes classification on the entire feature space (i.e., with no feature selection). This is to have a fair judgment in the presence of all the factors contributing to the CA, including both within-class variability (noise content of the signal) and between-class variability (dissimilarity of class wave-shapes). Figure 7a presents the Bayes CA in the salient feature space versus that of the same Bayes classifier in the original signal space. Hereafter referred to as the "CA–CA plot", the plot shown in Fig. 7a provides a sense of how the SFS method can improve the resilience of spike sorting against both within-class and between-class variabilities. The less-than-unity slope of the regression line (0.57) in the CA–CA plot of Fig. 7a indicates that (in addition to dimension reduction and consequently computational complexity reduction) the proposed SFS method makes the CA of spike sorting less sensitive to the aforementioned variabilities. Figure 7b compares the CA–CA plots of Bayes spike sorting when different approaches are taken for feature extraction/selection. According to this comparison, the proposed SFS method exhibits the most resilient CA against

signal variabilities (the smallest slope) and at the same time the highest CA.

To verify and evaluate the proposed spike sorting method, the sequence of forming the salient feature space followed by WD for neural spike classification is studied. The overall signal processing results of this method (SFS + WD) is compared with two other similar works that contain wave-shape classification. It should be noted that even though there are several works reporting on-implant spike sorting, the works of Karkare et al.[16] and Yang et al.[9] realize "complete" on-implant spike sorting (they go all the way to spike wave-shape classification as the very last step). In the former, an $l_1$-norm distance-based method is used for spike classification, which is referred to as the $l_1$-norm distance template matching ($l_1$-TM) for the classification of spike wave-shapes. As an alternative solution, the latter proposes the oblique decision tree (ODT) for on-implant spike sorting (Traditional classifiers such as Bayes have a high computational cost and therefore cannot be implemented on brain implants). Figure 8 compares the performance of the proposed method for 1- and 2-dimensional salient feature spaces (1D SFS + WD and 2D SFS + WD) with the other two approaches ($l_1$-TM and FBS_HT+ODT). In both 1D and 2D spaces, the SFS + WD method proves to be superior to the other techniques in terms of both CA and CA_CLI (i.e., with or without the influence of spike chance level) with reasonably small calculation times.

It was illustrated in Fig. 2b that the on-implant OSS comprises two main blocks: (I) The OSS internal parameters block, which consists of multiple register banks (holding the parameters received from the SSS), and (II) the OSS Engine, which mainly comprises simple digital comparators. The register banks in the OSS, which are shared among all the 512 channels[29], include

- a bank of a total of 5 k bits to contain salient feature indices,
- a bank of a total of 14 k bits to store the upper and lower bounds of the WDs for each and every salient feature, and
- a 3-k bit bank to hold the class identifiers associated with salient features.

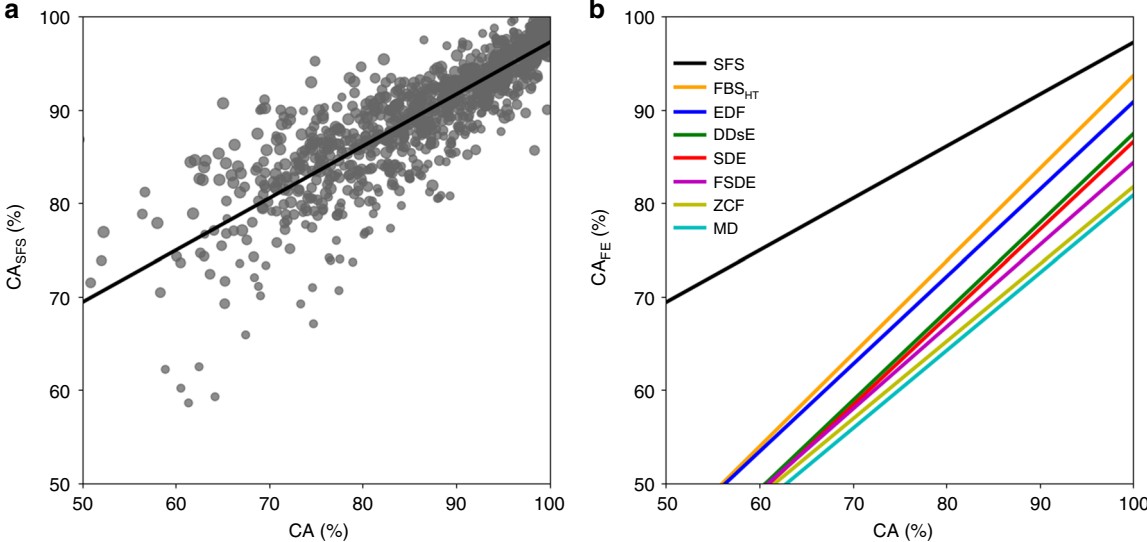

**Fig. 7 CA–CA plots. a** Bayes CA in salient feature space vs. Bayes CA without dimension reduction. **b** Regression lines for the feature extraction methods reported in Fig. 6. Given the fact that all the feature extraction/selection methods are followed by the same classifier, the CA–CA plot in **b** provides a fair basis for the comparison of the proposed salient feature strategy with other feature extraction/selection approaches. The top-most regression line in this plot belongs to the proposed approach. Exhibiting the highest possible classification accuracy ($CA_{FE|CA} = 100\%$) and at the same time the lowest slope among all approaches is an indication of the absolute superiority of the proposed salient feature selection idea over all other existing approaches in the presence of all sources of uncertainty (i.e., class similarity and between-class variability).

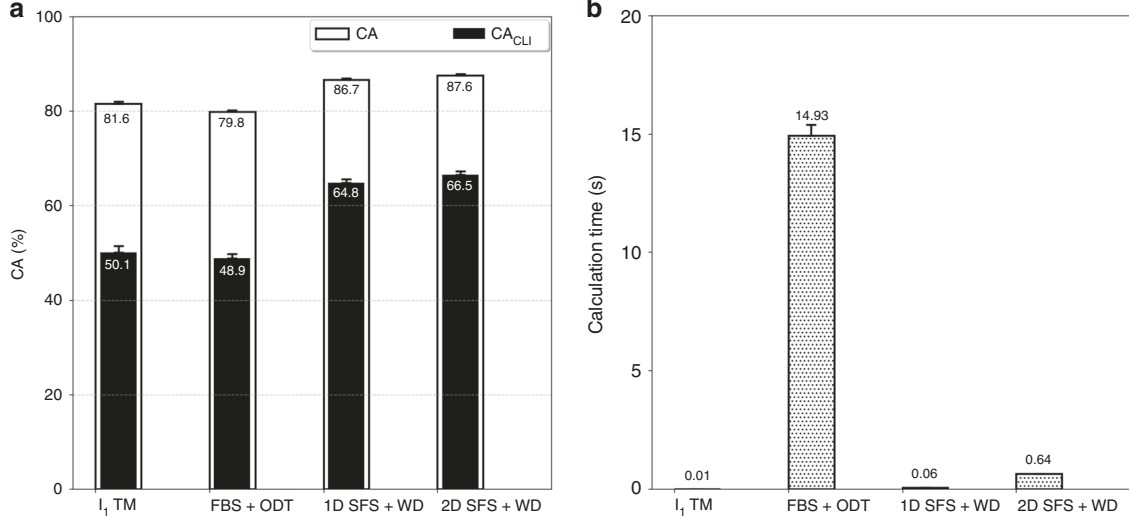

**Fig. 8 Comparison of the proposed spike sorting method with other similar works. a** CA and $CA_{CLI}$ for online spike sorting, **b** average calculation time for the training phase for all spike classes ($N = 1472$ independent spike classes including 125,120 test wave-shapes; data are presented as mean values ± SEM). The proposed spike sorting method offers meaningfully higher classification accuracy (both with and without the removal of the effect of the chance level) at a sufficiently low computation time.

The OSS engine is realized using merely a 5-bit comparator and two 7-bit comparators, which are properly time-shared among all the channels for WD tasks.

Compared with the works of Karkare et al.[16] and Yang et al.[9], the memory space required to implement the on-implant OSS proposed in this work is 5 times and 68 times smaller, respectively. In total, the on-implant O.S.S in this work is implemented using 1869 transistors per channel and takes a chip area of 0.0066 $\frac{mm^2}{ch.}$ in a 130-nm CMOS process. This is while the former work[16] and the latter work[9] occupy 0.077 $\frac{mm^2}{ch.}$ and 0.023 $\frac{mm^2}{ch.}$ in 65 and 130 nm CMOS technologies, respectively.

## Discussion

On-implant spike sorting methods normally use specific features with straightforward mathematical description to classify spike wave-shapes. Examples of such features are minima and maxima of the spike amplitude and their timing[3,11], maximum slopes (either positive or negative)[3–5,7,11], and zero-crossing times[12]. Even though those features correspond to critical points and important information of the spikes, but they are not necessarily the best possible features for spike wave-shape descrimination.

In this paper, we introduced a novel framework for on-implant spike sorting. The goal is to improve the CA and also reduce the

hardware cost. The proposed framework comprises the SFS method and WD for spike classification. The main aim of the SFS method is to efficiently reduce the dimension of data representation. The SFS method searches for the features that best distinguish each and every spike class from the rest of the spike classes in the signal space. It is guaranteed by definition that such features (referred to as "salient features") result in spike sorting in such a way that the geometric mean of between-class distances is maximized in the most homogeneous way. It is shown in this work that a set of such features can result in meaningfully higher classification accuracies compared with other spike sorting approaches existing in the literature (~2× reduction of classification error). The WD technique is used for multi-label classification of spike wave-shapes in the salient feature space. Taking advantage of both SFS and WD in a multi-label structure, online spike sorting is realized with higher CA at a significantly lower hardware cost (~5× reduction in the required memory), compared with other similar works reported.

In neural prosthetic applications, when activities of neural populations are monitored for long periods of time (hours, days, or weeks), although the number of units remains almost constant, the units might appear and disappear[30–32]. Such changes in the neural populations under study cause failure or at least degradation in the performance of the prosthesis. To handle and resolve such problems, the system in this work is designed to periodically recalculate the SFS and WD parameters (through the offline procedure already explained) and reconfigure the on-implant OSS accordingly. This results in maintaining the classification performance in the presence of such signal variations.

Taking into consideration physical and electrical limitations such as chip size and power consumption, a hardware prototype realizing the proposed spike sorting method is designed to be able to classify a total of 512 spike classes on all the 512 channels. One of the major practical requirements for the proposed spike sorting method is the physical size of its hardware implementation. To be mounted on the backside of a 100-channel Utah electrode array (with the area of ~4 × 4 mm2[33,34]), the silicon chip designed to realize the proposed method will therefore need to be smaller than ~16 mm2. Physical layout of the chip implementation of the proposed 512-channel spike sorter in a 130-nm standard CMOS technology occupies a silicon area of 3.36 mm2 (2.124 mm × 1.58 mm). Another physical concern in the development of a brain implant is the temperature increase it causes for the surrounding tissues. Temperature increase of more than 1–2 °C may damage the brain tissue[35–38], and therefore introduces a strict limitation on the power dissipated by the active circuitry on a brain implant. According to ref. [35], power density of a brain implant cannot exceed the upper limit of ~1.33 $\frac{mW}{mm^2}$ in order to keep the surrounding living tissues safe against temperature rise. Operated at a supply voltage of 1.2 V, the chip implementing the proposed spike sorting method dissipates a total power of 905.9 µW, which gives a safe power density of ~0.27 $\frac{mW}{mm^2}$.

As reported in Fig. 8, offline training time for the proposed 512-channel spike sorter (which is indeed the time required for the (re)configuration of the OSS) is ~5 min (0.64 s per channel). Even though this is somewhat larger than the configuration time reported in ref. [16], it is still significantly smaller than the much longer time (~30 min) that advanced brain machine interfaces typically require for (re)calibration (see refs. [30,31]).

## Methods

**Exponential class discrimination index.** As a measure for the normalized distance between the spike class under study (#i) and each one of the other spike classes (#j),

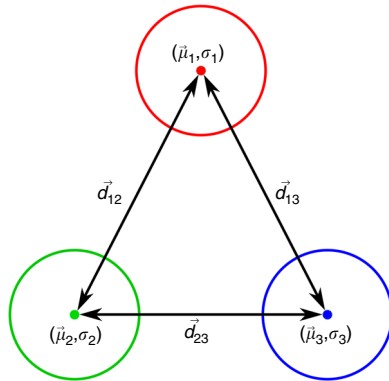

**Fig. 9 Illustration of class discrimination indices ($\vec{d}_{12}$, $\vec{d}_{13}$, and $\vec{d}_{23}$) for three spike classes in a two-dimensional feature space.** Each class is demonstrated by a circle, the center of which is the exemplar of the class obtained by sample-to-sample averaging of all the spikes in the class ($\mu_1$–$\mu_3$). "Within-class variability" (mainly caused by the random noise superposed on the spikes) and "between-class variability" stemming from the difference between the spike shapes of different classes are designated by $\sigma_i$ and $\vec{d}_{ij}$s, respectively.

it is proposed to use the exponential class discrimination index

$$d_{ij} = e^{\frac{|\mu_i - \mu_j|}{\sqrt{P_i \sigma_i^2 + P_j \sigma_j^2}}}, \qquad (5)$$

in which $(\mu_i, \mu_j)$, $(\sigma_i, \sigma_j)$, and $(P_i, P_j)$ are the mean values, standard deviations, and relative probabilities of occurrences for spike classes #i and #j, respectively. Figure 9 illustrates this distance measure in the case of three spike classes in a two-dimentional feature space.

**SFS in a K-dimensional feature space.** In order to come up with an optimum spike sorting solution, for each class a subset of L MSFs (out of the total of K features) is selected. This subset is referred to as the salient feature set for that class. To form the L-dimensional salient feature set for class #i, first, saliency of this class is calculated using each and every feature ($\varsigma_i[k]$, $1 \le k \le K$) as

$$\varsigma_i[k] = \frac{\left( \prod_{j=1(j \ne i)}^{N_c} (d_{ij}[k])^{P_j} \right)^2}{\sum_{j=1(j \ne i)}^{N_c} P_j \times d_{ij}[k]}, \qquad (6)$$

in which $d_{ij}[k]$ ($1 \le k \le K$) is the class discrimination index when classes are discriminated according to feature #k. Using the class saliency measure for each feature, the lth member of the salient feature set for class #i is determined as

$$k_i^l = \underset{\kappa \in \{1,2,...K\}}{\arg\max} \left\{ \varsigma_i[\kappa] \times \prod_{h=1}^{l-1} \left( 1 - \rho_i(\kappa, h) \right) \right\}, \qquad (7)$$

where $\rho_i(\kappa, h)$ indicates the correlation between the κth member of the main feature space and the hth member of the salient feature set (i.e., $k_i^h$).

**Reporting summary.** Further information on research design is available in the Nature Research Reporting Summary linked to this article.

## Data availability

The data set used for the current study are publicly available at http://crcns.org/data-sets/vc/pvc-1. The Python code used for analysis is available upon reasonable request.

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

## Author contributions

M.A.S. proposed the main idea, designed the work in both hardware and software, and drafted the paper; A.M.S. was the academic supervisor of the work and substantially revised the paper; and both authors analyzed and interpreted the results.

## Competing interests

The authors declare no competing interests.
