## [Peer Review File · Nature Communications]

Reviewers' Comments:

Reviewer #1:

Remarks to the Author:

The article: "On-Implant Online Spike Sorting based on Salient Feature Selection" describes the novel framework for real-time online spike sorting on implantable neural recording micro-systems.

The authors report the novel on-implant spike sorting method, which w.r.t. other methods employing static feature selection/extraction techniques to minimize the hardware cost, is based on a dynamic selection of features.

As the main result, they concluded that compared with other similar works, this method shows reduction in classification error by a factor of ~ 2 (as the improvement in signal processing performance) and also reduction in the required memory space by a factor of ~ 5 (as the reduction in hardware implementation cost).

The key value of the proposed spike sorting technique is in its potential to allow for a power and area-efficient hardware implementation that operates in the real time on a high-density neural recording implant.

I find their approach very interesting and of the possible usability in the future brain-machine interfacing.

Yet I have few major comments, which should help to better understand the proposed system, with all its limitations and pros.

My main concern is putting the present, very specifically technological article, into the wider context, so understandable also to the non-experts from the BMI or spike sorting area. To do so, an effort should be placed in adding figure, which is depicting where and what is done w.r.t. brain implant, skull etc. Also the wording should be heavily re-worked. It was extremely time consuming and required a significant effort to follow-up on the steps of method implementation. The datasets used and their division in training and testing are not very clear. Significant parts of sections 1.2. and 1.3. could be possibly placed in the supplementary methods.

Comparison with the real BMI systems (e.g. Schwartz, Donoghue, Carmena, Fetz...) should be given, and these works should be cited.

The statical methods used are not detailed, and should be provided.

It is not very clear if the real system (device) having the structure from the schematic Fig.1. has been implemented. The section 2.3. Hardware Implementation does not help to understand if this has been done in the present work, or in some other paper, and how to physically replicate that device construction. If yes, the performances of that system should be reported, at least from the benchmark testing. If not, the clear indications about how it will be done, starting from the present method, should be provided.

Authors state: "reduction in classification error by a factor of ~ 2 (as the improvement in signal processing performance)..."

This is a very strong statement and should be better explained, since it is hard to understand what do they mean by doubling the classification precision? Which aspect of it?

Reviewer appreciates this section: "It should be added that in neural prosthetic applications, when activities of neural populations are monitored for long periods of time (hours, days, or weeks),

although the number of units remains almost constant, the units might appear and disappear [29–31]. Such changes in the neural populations under study cause failure or at least degradation in the performance of the prosthesis. To handle and resolve such problems, the system in this work is designed to periodically recalculate the SFS and WD parameters (through the offline procedure already explained) and reconfigure the on-implant O.S.S accordingly. This results in maintaining the classification performance in the presence of such signal variations.”

Then the logical question that arises is if the authors did do this work with the experimental datasets they used? Is their system able to compensate for this spike alterations, which are inherent to all neuroprosthetic applications?

Limitations of a present approach should be clearly stated in the discussion section.

The quality of figures should be higher, and some of figures (e.g. 6,8) could become the panels of the same figure.

Reviewer #2:

Remarks to the Author:

In this work, Shaeri and Sodagar describe an approach for online spike sorting directly on brain implants. This is a difficult problem to solve under the typical power and space constraints of on-implant chips, and one that is timely to solve now, given the recent wave of technological advancements in CMOS electrode fabrication. The authors propose a sensible approach of using a powerful offline system for finding the most informative features for discriminating neurons, thereby reducing the online computational load on the implant. However, I should point out that I do not have the expertise to validate the feasibility claims for the hardware solution, and the authors do not actually produce such a device to show that it works in practice.

I don't think the paper requires any new major analyses, but I think the presentation should be substantially streamlined to better communicate the results. For example, the figure captions could be longer and more descriptive, and contain fewer acronyms not defined in the caption (ideally none). The entire text would benefit from much less use of acronyms, especially because these are not general acronyms, and defined by the authors for this paper only.

Some comment in the "Results" section should be moved to the discussion. For example the paragraph about periodic re-optimization of the system, which is not quantified or implemented. Most of the definitions in the "Basic concepts" section should be moved to methods, and introduced in the text where they are first used, with a single explanatory sentence. This is already done for the "Salient features selection".

The presentation in figure 6 is confusing: the number of features should be encoded as text on the bars, or on the horizontal axis, not as a secondary plot axis. I would also like to see the proposed method with more than two features, since any improvements above 89% accuracy would be welcome and important. I see a similar problem in figure 8. I think this one could split into two bar plots, instead of using the same axis.

Finally, I hope that the authors will explore in the future the feasibility of on implant multi-channel spike sorting, to take advantage of the higher density of standard silicon probes and the recent CMOS probes. Also, coordinating with groups that fabricate such probes might enable faster translation of the technology described in this paper.

Marius Pachitariu

Reviewer #3:

Remarks to the Author:

Reviewers Comments: On-Implant Online Spike Sorting based on Salient Feature Selection, Nature Communications, NCOMMS-19-15270, by Mohammad Ali Shaeri and Amir M. Sodagar.

These authors report a method for spike sorting intra-cranial neural data for imbedded, wireless recording and feedback devices, such as robot arm control or other prosthetic applications. They argue that their encoding strategy published earlier (their reference s) combined with a "salient feature selection" strategy for clustering give both superior performance and can be embodied in hardware with smaller footprint and lower power consumption than the two best (their reference) currently available alternatives, their references 9 and 16. While the authors never specifically state the data for which this strategy is intended, they test against a public database of waveforms recorded with single wire electrodes in rhesus macaque. I will assume this data type is the intended target or some future human application with equivalent recording technology. The manuscript seems to be technically correct and the figures are effective. I note a few more typographical issue below.

The accuracy illustrated in Figure 6 and 7 support the advantage of the salient feature method seems to be a substantial improvement over the two competing methods. Figure 8 shows modest increases in accuracy over both other methods and significant reduction in calculation time over the slower of the two (the figure labels are "Clac. Time", I assume this is meant to be Calc. Time). My concern is significance. As far as I can tell, the authors assume an input to their feature extractor/clustering system of 512 channels remote digitized recording signals. The significance concern arises from two directions. First, calling these data spike sortable and these methods spike sorting would have no meaning in the rodent electrophysiology community. Harris et al. J Neurophysiology 2000, showed that single channel data are seriously error prone to spike assignment compared to tetrode recording results from the same data using all available waveform information. On top of this inherent error, a very recent paper of high channel count data from dense arrays in macaque motor cortex show no advantage of spike sorted data over spike occurrence per channel in model limb trajectories (Trautmann et al. Neuron 2019). A simple spike threshold detector per channel would be far simpler and power efficient than this proposed method and apparently comparably accurate for motor prediction for motor cortex. So, would this method have any significance even if fully implemented. Second is the assumption of the available remote input. The smallest commercially available integrated circuit to amplify and digitize signals is the Intan Technologies RHD2164 bare die 64 channel chip, which is 7.3x4.2 mm per 64 channels, or 245 mm² for 512 channels. Alternatively, the Neuropixels probe, Jun et al., Nature 2017, reports an area of ~54 mm² for 384 channels, or ~72 mm² for 512 channels. The authors claim an area of 3.4 mm² needed for spike sorting while the prior best art needs 11.8 mm² (authors ref. 9) or 39.4 mm² (authors reference 16). The latter, while much larger is 6X faster (figure 8). Comparing to the available 245 mm² input requirement, the improvement (~25% more accurate spike assignment but still only ~65%) while 6X slower. I conclude that while there is engineering validity to the approach, the application value is lacking. Real but modest improvement in accuracy, significant reduction in chip area, but input chip area requirements dwarf the chip area saving. Add this to the apparent absence of clear value for spike sorting in predicting trajectories does not predict a user base for this strategy.

Finally, I note to the authors that the data set used to test the algorithms is unclear from the description. The authors state they use a data base of ~15,000 waveforms but say nothing about the realism of their sample. Are the 3 waveforms chosen for each channel for the same brain area? How

are these chosen waveforms varied to create a clustering sample. The text reads as if assignments are made for 1 example of the 2-4 waveforms for each chosen for a channel. Do they really call classifying ~1450 spikes, 2-4 spikes per computation, clustering? What is the equivalent length of the recording and the per spike classification time? If this is a prosthetic application, what time is allowed for classification? Unfortunately this method as with all other spike sorting methods suffers from a lack of good ground truth data that scales with the real problem. I do understand how these authors have addressed that problem.

At the top of page 5 the authors repeat the phrase "according to the definition" twice.

Dear Reviewers,

The authors would like to thank the Reviewers for their helpful comments, which helped improve the strength and clarity of the paper. What follows is a complete list of the reviewers' comments (in black), and the authors' responses in blue under each comment.

The revised version being submitted has been prepared with (and without) the following color codes:

- black: remains as it is,
- red: added/modified according to the reviewers' comments,
- gray: deleted (if moved to a different section of the manuscript, it appears in the new location in red.).

Reviewer #1

The article: "On-Implant Online Spike Sorting based on Salient Feature Selection" describes the novel framework for real-time online spike sorting on implantable neural recording micro-systems.

The authors report the novel on-implant spike sorting method, which w.r.t. other methods employing static feature selection/extraction techniques to minimize the hardware cost, is based on a dynamic selection of features.

As the main result, they concluded that compared with other similar works, this method shows reduction in classification error by a factor of ~ 2 (as the improvement in signal processing performance) and also reduction in the required memory space by a factor of ~ 5 (as the reduction in hardware implementation cost).

The key value of the proposed spike sorting technique is in its potential to allow for a power and area-efficient hardware implementation that operates in the real time on a high-density neural recording implant.

I find their approach very interesting and of the possible usability in the future brain-machine interfacing.

Yet I have few major comments, which should help to better understand the proposed system, with all its limitations and pros.

My main concern is putting the present, very specifically technological article, into the wider context, so understandable also to the non-experts from the BMI or spike sorting area.

- 1. To do so, an effort should be placed in adding figure, which is depicting where and what is done w.r.t. brain implant, skull etc.*

The authors would like to thank the reviewer for this comment. Figure 1 is now added in order to illustrate placement of the implant module as well as its wireless bidirectional communication with the external setup.

- 2. Also the wording should be heavily re-worked. It was extremely time consuming and required a significant effort to follow-up on the steps of method implementation.*

In order to implement this useful comment,

- For a better description of how the system works, a new paragraph was added at the end of Section 1 to clearly introduce the key parts of the system and also their general functions.
- Moreover, the first paragraph of Section 2 was edited for more clarity.
- Also, some more explanation was added after that (first 6 lines of the second paragraph) in order to explain the steps of method implementation as per this reviewer's comment.

3. *The datasets used and their division in training and testing are not very clear.*

The reviewer is right. Section 3 of the revised manuscript now contains more details of the datasets used for training and testing.

4. *Significant parts of sections 1.2. and 1.3. could be possibly placed in the supplementary methods.*

Arrangement of the manuscript has now changed in order to implement this comment.

5. *Comparison with the real BMI systems (e.g. Schwartz, Donoghue, Carmena, Fetz...) should be given, and these works should be cited.*

Most of existing BMIs are bench-top systems. As examples, one can point to Schwartz's work in [ref 30: Collinger,'13][Inoue'18], Donoghue's work in [Ajiboye'17], Carmena's work in [Zhou'19], Fetz's work in [Moritz'08] Hochberg's work in [ref 29: hochberg'12], and Andersen's work in [ref 31:Aflalo'15]. The bench-top part of such systems is in charge of data acquisition from the brain (in a hardwired manner) and also online signal processing (including spike sorting). Signal processing in this category of BMIs can, therefore, be implemented on high-performance computers in the external setup with virtually no restriction in power consumption or physical dimensions.

Next-generation neural interfacing systems are fully-implantable high-density brain-implantable micro-systems. Even though there are reports in the literature on the development and test of such systems (e.g., [Sodagar'09]), for Brain-Machine Interfacing (BMI) applications, real-time operation on at least hundreds of channels is essential. With the recent advancement in the fabrication of high-density microelectrode arrays (e.g., [Jun'2017]), major technical bottleneck in making complete high-density fully-implantable BMI systems is now the transfer of a huge amount of recorded data off the implant. As an efficient solution, suggestion of sufficiently-accurate spike sorting methods that can be efficiently* implemented on implantable micro-systems is of crucial importance. As explained and cited in the manuscript, to the best of the authors' knowledge, there are two pioneering works in the literature focusing on this specific matter (references [16] and [9] of the manuscript). To be fair and in similar implementation conditions, our results are, therefore, compared with those of those two works.

**from the standpoint of computational complexity, hardware complexity, power consumption, and operation in real time*

[Maharbiz'17] Maharbiz, M., M., Muller, R., Alon, E., Rabaey, J. M. & Carmena, J. M. Reliable Next-Generation Cortical Interfaces for Chronic Brain–Machine Interfaces and Neuroscience. *Proceedings of the IEEE* **105**, 73-82 (2017).

[Brandman'17] Brandman, D. M., Cash, S. S. & Hochberg, L. R. Review: Human Intracortical Recording and Neural Decoding for Brain–Computer Interfaces. *IEEE Transactions on Neural Systems and Rehabilitation Engineering* **25**, 1687–1696 (2017).

[Ajiboye'17] Ajiboye, A. B., et al, Restoration of reaching and grasping movements through brain-controlled muscle stimulation in a person with tetraplegia: a proof-of-concept demonstration. *Lancet* **389**, 1821-1829 (2017).

[Inoue'18] Inoue, Y., Mao, H., Suway, S.B., Orellana, J. and Schwartz, A.B., 2018. Decoding arm speed during reaching. *Nature communications* **9**, 1-18 (2018).

[Zhou'19] Zhou, A., et al, A wireless and artefact-free 128-channel neuromodulation device for closed-loop stimulation and recording in non-human primates. *Nature biomedical engineering* **3**, 15-26 (2019).

[Moritz'18] Moritz, C.T., Perlmutter, S.I. and Fetz, E.E. Direct control of paralysed muscles by cortical neurons. *Nature* **456**, 639-643 (2008).

[Sodagar'09] Sodagar, A. M., Perlin, G. E., Yao, Y., Najafi, K. & Wise, K. D. An implantable 64-channel wireless microsystem for single-unit neural recording. *IEEE Journal of Solid-State Circuits* **44**, 2591–2604 (2009).

6. *The statical methods used are not detailed, and should be provided.*

The authors' guess is that by “statical methods”, the reviewer means “static feature selection/extraction methods”, referring to the sentence “While other on-implant spike sorting methods employ static feature selection/extraction techniques to ...” in the Abstract. As clearly mentioned there, this is the work of other groups, described in the first paragraph of section 3.1 in the revised manuscript. To clarify the term “static” as per reviewer’s comment, more explanation is now added there.

7. *It is not very clear if the real system (device) having the structure from the schematic Fig.1. has been implemented. The section 2.3. Hardware Implementation does not help to understand if this has been done in the present work, or in some other paper, and how to physically replicate that device construction. If yes, the performances of that system should be reported, at least from the benchmark testing. If not, the clear indications about how it will be done, starting from the present method, should be provided.*

Focus of this comment is on two issues: (a) Implementation, and (b) Performance.

(a) IMPLEMENTATION: As explained in the second paragraph of our response to Comment #5, main focus of this work (and similar works being reported in the literature) is to provide an

efficient solution for “on-implant online spike sorting” (as the title of the manuscript states) with implant-appropriate hardware implementation. The part of the system in charge of the implementation of the proposed idea has been completely designed and implemented using standard digital hardware. Figure 2.b in the revised manuscript shows more details of this implementation. Since the main focus of this paper is on the introduction of a new spike sorting method, it intentionally elaborates on the details of the classification method itself. The authors have also provided some information on the implant-appropriate hardware implementation design and performance. However, details of the engineering design of the silicon chip and its verification and performance tests take an extensive journal publication, the authors have the plan to publish in the near future.

- (b) PERFORMANCE: All the spike sorting results presented in Section 3 have been achieved by taking into account hardware digital implementation requirements. More specifically, subsection 3.3 focuses on electrical and physical details of the hardware design. On the signal processing side, we have
- reported classification accuracies,
 - introduced a new measure for classification accuracy independent from chance levels (CA_{CLI}), and have provided CA_{CLI} values (not only for our work, but also for other similar works using the same data sets),
 - presented CA-CA plots exhibiting resilience of CA of our proposed method against signal variabilities (Figure 7.b in the revised manuscript).

From the hardware performance standpoint, we have reported the following information:

- hardware design details (memory space allocations, computational blocks, etc.),
- per-channel transistor count, and
- silicon area,
- timing of the training phase, and
- timing of the online spike sorting phase.

8. *Authors state: “reduction in classification error by a factor of ~2 (as the improvement in signal processing performance)..”. This is a very strong statement and should be better explained, since it is hard to understand what do they mean by doubling the classification precision? Which aspect of it?*

As shown in Figure 6 of the revised manuscript (identical to Figure 6 in the original submission), classification accuracies CA and CA_{CLI} for the proposed method (SFS) are 89.5% and 73.4%, respectively, corresponding to classification errors of 10.5% and 26.6%. For the most successful work of others (FBS_{HT}), the same classification accuracies are 80.6% and 50.4%, respectively, meaning that the associated classification errors are 19.4% and 49.6%. Comparing the errors, the reductions achieved in classification errors in the regular way and after the removal of chance levels are 10.5/19.4 (~0.54) and 26.6/49.6 (~0.53), respectively. We have claimed that the classification errors are approximately halved, which definitely does not mean (and the authors have never claimed) that the precision has been doubled.

9. Reviewer appreciates this section: “It should be added that in neural prosthetic applications, when activities of neural populations are monitored for long periods of time (hours, days, or weeks), although the number of units remains almost constant, the units might appear and disappear [29–31]. Such changes in the neural populations under study cause failure or at least degradation in the performance of the prosthesis. To handle and resolve such problems, the system in this work is designed to periodically recalculate the SFS and WD parameters (through the offline procedure already explained) and reconfigure the on-implant O.S.S accordingly. This results in maintaining the classification performance in the presence of such signal variations.” Then the logical question that arises is if the authors did do this work with the experimental datasets they used? Is their system able to compensate for this spike alterations, which are inherent to all neuroprosthetic applications?

Yes, as explained in the manuscript and the reviewer has quoted, “To handle and resolve such problems, the system in this work is designed to periodically recalculate the SFS and WD parameters (through the offline procedure already explained) and reconfigure the on-implant O.S.S accordingly.” It should be added that from engineering design perspective, continually monitoring all the channels and checking for the presence of all the units the system is trained for is impractical, as it is extremely time consuming, and does not allow for real-time operation of the spike sorter.

10. Limitations of a present approach should be clearly stated in the discussion section.

Thanks to this comment, the following section is added to the revised manuscript:

4. Discussion

Taking into consideration physical and electrical limitations such as chip size and power consumption, the hardware prototype realizing the proposed spike sorting method is designed to be able to classify a total of 512 spike classes on all the 512 channels.

One of the major practical requirements for the proposed spike sorting method is the physical size of its hardware implementation. To be mounted on the backside of a 100-channel Utah electrode array (with the area of $\sim 4 \times 4 \text{ mm}^2$ [33, 34]), the silicon chip designed to realize the proposed method will therefore need to be smaller than 16 mm^2 . Physical layout of the chip implementation of the proposed 512-channel spike sorter in a 130 nm standard CMOS technology occupies a silicon area of 3.36 mm^2 ($2.124 \text{ mm} \times 1.58 \text{ mm}$).

Another physical concern in the development of a brain implant is the temperature increase it causes for the surrounding tissues. Temperature increase of more than 1-2 °C may damage the brain tissue [35–38], and therefore introduces a strict limitation on the power dissipated by the active circuitry on a brain implant. According to [35], power density of a brain implant cannot exceed the upper limit of 1.33 mW/mm^2 in order to keep the surrounding living tissues safe against temperature rise. Operated at a supply voltage of 1.2V, the chip implementing the proposed spike sorting method dissipates a total power of $905.9 \mu\text{W}$, which gives a safe power density of $\sim 0.27 \text{ mW/mm}^2$.

As reported in Figure 8, offline training time for the proposed 512-channel spike sorter (which is indeed the time required for the (re)configuration of the online spike sorter) is ~ 5 minutes (0.64s per channel). Even though this is somewhat larger than the configuration time reported in

[16], it is still significantly smaller than the much longer time (~30 minutes) that advanced BMI systems typically require for (re)calibration (see [29], [30]).

[Authors: The following passage is added for reviewers' information. The authors do not think this much of details falls in the scope of this paper. However, if the authors or the Associate Editor believe that it is better be included in Discussions, we will do so in the final version.]

From the perspective of the bit rate utilized by the system, according to the block diagram presented in Figure 2, there are four major streams of digital data exchanged with the proposed online spike sorter:

- (i) With a sampling rate of 20ksps and resolution of 8 bits, spike wave-shapes in the digitized raw input neural signals on all the 512 channels enter the online spike sorter with a maximum rate of 2.56 Mbps,
- (ii) for the training phase, streaming of neural signals from the implant to the external module is performed with the same data rate as in (i),
- (iii) the total configuration data calculated on the external module is ~27.65 kbits. With a bit rate of 2.56 Mbps, it takes ~.01 sec. to telemeter this data from the external module to the implant, and
- (iv) sorted spike information on all the 512 channels is sent off the implant in the form of a binary spike train stream with a total bit rate of 512 kbps.

11. *The quality of figures should be higher, and some of figures (e.g. 6,8) could become the panels of the same figure.*

Figures 5, 6, and 8 in the revised manuscript are reproduced with better quality.

Reviewer #2

In this work, Shaeri and Sodagar describe an approach for online spike sorting directly on brain implants. This is a difficult problem to solve under the typical power and space constraints of on-implant chips, and one that is timely to solve now, given the recent wave of technological advancements in CMOS electrode fabrication. The authors propose a sensible approach of using a powerful offline system for finding the most informative features for discriminating neurons, thereby reducing the online computational load on the implant.

12. *However, I should point out that I do not have the expertise to validate the feasibility claims for the hardware solution, and the authors do not actually produce such a device to show that it works in practice.*

The authors would like to refer the reviewer to part (a) of their response to Comment #7.

13. *I don't think the paper requires any new major analyses, but I think the presentation should be substantially streamlined to better communicate the results. For example, the figure captions could be longer and more descriptive, and contain fewer acronyms not defined in the caption (ideally none). The entire text would benefit from much less use of acronyms, especially because these are not general acronyms, and defined by the authors for this paper only.*

The authors thank the reviewer for bringing this issue to their attention. The acronyms are omitted from the caption of Figure 2. Also, the following acronyms are removed from the text: Implantable module (IM), External module (EM), Feature selection (FS), Exponential class discrimination index (ECDI) and Gaussian Naive Bayes (GNB).

14. *Some comment in the "Results" section should be moved to the discussion. For example the paragraph about periodic re-optimization of the system, which is not quantified or implemented. Most of the definitions in the "Basic concepts" section should be moved to methods, and introduced in the text where they are first used, with a single explanatory sentence. This is already done for the "Salient features selection".*

Arrangement of the manuscript has now changed in order to implement this comment.

15. *The presentation in figure 6 is confusing: the number of features should be encoded as text on the bars, or on the horizontal axis, not as a secondary plot axis.*

Number of features is now put on the top of each hashed bar.

16. *I would also like to see the proposed method with more than two features, since any improvements above 89% accuracy would be welcome and important.*

The choice of the number of salient features was based on the optimization of the classification task considering both the accuracy achieved and the time spent for calculations. If the number of features for the SFS method is changed from 2 to 3, classification accuracy increases from 89.5% ($CA_{CLI}=73.4\%$) [in Figure 6] to 90.2% ($CA_{CLI}=75.1\%$). The cost being paid for this slight increase in the accuracy, however, is the additional calculation time. The fact of the matter is that the calculation time (for training phase, i.e. estimation of SFS+WD parameters) increases with an exponential pace (with respect to the number of features) from .64s to 9.9s. Therefore, with a straightforward cost-benefit analysis, it was decided to go with two salient features.

17. *I see a similar problem in figure 8. I think this one could split into two bar plots, instead of using the same axis.*

The plot in Figure 8 is now split into two separate plots as per this comment.

18. *Finally, I hope that the authors will explore in the future the feasibility of on implant multi-channel spike sorting, to take advantage of the higher density of standard silicon probes and the recent CMOS probes. Also, coordinating with groups that fabricate such probes might enable faster translation of the technology described in this paper.*

In complete agreement with the reviewer, as it is the common practice in such works, the plan is to employ this method in a complete high-density neural recording system and test it in real conditions. The authors would like to thank the reviewer for providing such useful comments.

These authors report a method for spike sorting intra-cranial neural data for imbedded, wireless recording and feedback devices, such as robot arm control or other prosthetic applications. They argue that their encoding strategy published earlier (their reference s) combined with a “salient feature selection” strategy for clustering give both superior performance and can be embodied in hardware with smaller footprint and lower power consumption that the two best (their reference) currently available alternatives, their references 9 and 16.

The authors would like to express their gratitude to the reviewer for providing the following useful comments, which helped clarify some issues in the revised manuscript.

19. *While the authors never specifically state the data for which this strategy is intended, they test against a public database of waveforms recorded with single wire electrodes in rhesus macaque.*

The data set cited in the ‘Results’ section of the first submission was recorded using a 10x10 Utah microelectrode array. This is now clarified in Section 3 of the revised manuscript. More details on the data set, and also how it is used for training and tests has also been added to that section.

I will assume this data type is the intended target or some future human application with equivalent recording technology. The manuscript seems to be technically correct and the figures are effective. I note a few more typographical issue below.

20. *The accuracy illustrated in Figure 6 and 7 support the advantage of the salient feature method seems to be a substantial improvement over the two competing methods. Figure 8 shows modest increases in accuracy over both other methods and significant reduction in calculation time over the slower of the two (the figure labels are “Clac. Time”, I assume this is meant to be Calc. Time).*

As for comparing classification accuracies in Figure 8 of the revised manuscript, the authors would like to bring to the reviewer’s attention the importance of eliminating the impact of chance levels from the accuracies achieved in spike classification. As suggested in section 2.1, CA_{CLI} is defined in this paper to measure the success of spike classification techniques independently from the chance of firing for the units present in the test data. Therefore, taking CA_{CLI} as a more fair comparison criterion to judge on the success of the techniques, the proposed method is at least $(64.8-50.1)/50.1 \approx 30\%$ more accurate than the other techniques it is compared with. However, advantages of the proposed method over its two competitors (i.e., [16], [9]) are not limited to only higher classification accuracy. For instance, being specifically dedicated to brain implants, silicon area of the hardware implementations is of crucial importance. As reported in “Results”, silicon area per channel in our design is significantly lower than the others’: 3.5 times smaller than the design reported in [9], and 46.7 times smaller than the work in [16] (when projected from their 65nm technology to a 130 nm technology for fair comparison).

The authors thank the reviewer for catching this typo. “Clac. Time” is now corrected to “Calculation Time”

21. My concern is significance. As far as I can tell, the authors assume an input to their feature extractor/clustering system of 512 channels remote digitized recording signals. The significance concern arises from two directions:

First, calling these data spike sortable and these methods spike sorting would have no meaning in the rodent electrophysiology community. Harris et al. *J Neurophysiology* 2000, showed that single channel data are seriously error prone to spike assignment compared to tetrode recording results from the same data using all available waveform information. On top of this inherent error, a very recent paper of high channel count data from dense arrays in macaque motor cortex show no advantage of spike sorted data over spike occurrence per channel in model limb trajectories (Trautmann et al. *Neuron* 2019). A simple spike threshold detector per channel would be far simpler and power efficient than this proposed method and apparently comparably accurate for motor prediction for motor cortex. So, would this method have any significance even if fully implemented.

With all due respect to the reviewer’s opinion, the authors do not believe that the paper by Harris et al. in *J Neurophysiology* 2000 disqualifies/rejects the usefulness of spike sorting. It even does not seem to claim that spike wave shape information does not add to the ultimate quality of neural decoding. The authors admit that there is a debate in the neural decoding community on the usefulness/effectiveness of spike sorting. However, there are several new works published in prestigious journals, reporting neural decoding on different animal subjects (rodents and monkeys), published even after Trautmann et al. *Neuron* 2019. Examples of such publications are:

- Mohan, H., de Haan, R., Broersen, R., Pieneman, A.W., Helmchen, F., Staiger, J.F., Mansvelder, H.D. and de Kock, C.P. Functional architecture and encoding of tactile sensorimotor behavior in rat posterior parietal cortex. *Journal of Neuroscience* **39**, 7332-7343 (2019).
- Laboy-Juárez, K.J., Langberg, T., Ahn, S. and Feldman, D.E. Elementary motion sequence detectors in whisker somatosensory cortex. *Nature neuroscience* **22**, 1438-1449 (2019).
- Wood, K.C., Town, S.M. and Bizley, J.K. Neurons in primary auditory cortex represent sound source location in a cue-invariant manner. *Nature communications* **10**, 1-15 (2019).
- Stringer, C., Pachitariu, M., Steinmetz, N., Carandini, M. and Harris, K.D. High-dimensional geometry of population responses in visual cortex. *Nature* **571**, 361-365 (2019).
- Lemke, S.M., Ramanathan, D.S., Guo, L., Won, S.J. and Ganguly, K., 2019. Emergent modular neural control drives coordinated motor actions. *Nature neuroscience* **22**, 1122-1131 (2019)
- Miyamoto, H., Tatsukawa, T., Shimohata, A., Yamagata, T., Suzuki, T., Amano, K., Mazaki, E., Raveau, M., Ogiwara, I., Oba-Asaka, A. and Hensch, T.K. Impaired cortico-striatal excitatory transmission triggers epilepsy. *Nature communications* **10**, 1917 (2019).
- Kells, P.A., Gautam, S.H., Fakhraei, L., Li, J. and Shew, W.L. Strong neuron-to-body coupling implies weak neuron-to-neuron coupling in motor cortex. *Nature communications*, **10**, 1575 (2019).

- o Shahidi, N., Andrei, A.R., Hu, M. and Dragoi, V., 2019. High-order coordination of cortical spiking activity modulates perceptual accuracy. *Nature neuroscience* **22**, 1148–1158 (2019).

22. *Second is the assumption of the available remote input. The smallest commercially available integrated circuit to amplify and digitize signals is the Intan Technologies RHD2164 bare die 64 channel chip, which is 7.3x4.2 mm per 64 channels, or 245 mm² for 512 channels. Alternatively, the Neuropixels probe, Jun et al., Nature 2017, reports an area of ~54 mm² for 384 channels, or ~72 mm² for 512 channels. The authors claim an area of 3.4 mm² needed for spike sorting while the prior best art needs 11.8 mm² (authors ref. 9) or 39.4 mm² (authors reference 16). The latter, while much larger is 6X faster (figure 8). Comparing to the available 245 mm² input requirement, the improvement (~25% more accurate spike assignment but still only ~65%) while 6X slower. I conclude that while there is engineering validity to the approach, the application value is lacking. Real but modest improvement in accuracy, significant reduction is chip area, but input chip area requirements dwarf the chip area saving. Add this to the apparent absence of clear value for spike sorting in predicting trajectories does not predict a user base for this strategy.*

As the reviewer knows, the Intan RHD 2164 is a commercial chip for pre-amplification, band-pass filtering, and digitization of neural signals. As shown in the data sheet of this IC, part of the chip area is occupied for the realization of a controller to realize the standard SPI protocol for interfacing with computerized/embedded systems and/or commercial lab equipment. Moreover, some area is used for a pad frame all around the chip in order to allow for the on-chip circuitry to be wire-bonded to a standard package (or directly bonded to a custom-made platform). This is while when it comes to the development of fully-implantable neural recording microsystems, integration of the recording front-end and spike sorting modules on the same chip will eliminate the need for neither standard communication protocols nor connection through bonding pads. Another key issue that determines the chip area is the microfabrication technology/process, which directly scales the dimensions of the design. No information is provided in the data sheet of this chip, but from the supply voltage of 3.3V, it is evident that the microfabrication technology of this chip is rather old with a “minimum feature size” of 0.35µm or larger. As a natural conclusion, the area of the same circuit would be much smaller if it had been designed with today’s processes (e.g., 130nm, 90nm, or 65nm).

For a summary of examples of area-efficient neural recording front-ends, the authors would like to refer the reviewer to Table I of [Luo’19], and Table I of [Rezaei’18]. Therefore, the signal preconditioning block can be as small as 0.013mm²/channel, giving an estimated chip area of ~6.66mm² for signal preconditioning on a total of 512 channels. As reported in the manuscript, the idea presented in our work is realized with a total chip area of 3.36mm² for 512 channels, which puts its application value into perspective. As explained in our response to Comment-20, it is reported in “Results” that silicon area per channel in our design is significantly lower than the others’: 3.5 times smaller than the design reported in [9], and 46.7 times smaller than the work in [16] (when projected from their 65nm technology to a 130 nm technology for fair comparison).

As for the values of the proposed technique in terms of accuracy and calculation time, the authors would like to refer the reviewer to their responses to comments 8 and 20.

[Luo'19] Luo, D., Zhang, M. and Wang, Z. A Low-Noise Chopper Amplifier Designed for Multi-Channel Neural Signal Acquisition. *IEEE Journal of Solid-State Circuits* **54**, 2255-2265 (2019).

[Rezaei'18] Rezaei, M., Maghsoudloo, E., Bories, C., De Koninck, Y. and Gosselin, B. A low-power current-reuse analog front-end for high-density neural recording implants. *IEEE transactions on biomedical circuits and systems* **12**, 271-280 (2018).

23. Finally, I note to the authors that

- *the data set used to test the algorithms is unclear from the description. The authors state they use a data base of ~15,000 waveforms but say nothing about the realism of their sample.*

Extensive clarification and more explanation on the data set was added to Section 3 “Results” of the revised manuscript.

- *Are the 3 waveforms chosen for each channel for the same brain area?*

Yes, they are all recorded from primary visual cortex (V1) using the same electrode array.

- *How are these chosen waveforms varied to create a clustering sample. The text reads as if assignments are made for 1 example of the 2-4 waveforms chosen for a channel. Do they really call classifying ~1450 spikes, 2-4 spikes per computation, clustering?*

The authors are grateful to the reviewer for bringing this issue to our attention. It's ~1450 spike classes (unit activities). Section 3 “Results” has now been edited and contains more clarification on the data used for training and tests.

- *What is the equivalent length of the recording and the per spike classification time? If this is a prosthetic application, what time is allowed for classification?*

According to the information provided in [Nauhaus'07] on their neuroscience tasks, each trial of our data set takes at least 240 seconds. As for per-spike classification time, thanks to its light computational complexity, the proposed on-implant online spike sorting process is fast enough to be completed during the course of the spike. The associated class label is therefore available at the output as soon as the spike ends.

Nauhaus, I., Ringach, D. L. Precise Alignment of Micromachined Electrode Arrays With V1 Functional Maps. *Journal of Neurophysiology* **97**, 3781-3789 (2007).

- *Unfortunately this method as with all other spike sorting methods suffers from a lack of good ground truth data that scales with the real problem. I do understand how these authors have addressed that problem.*

- *At the top of page 5 the authors repeat the phrase “according to the definition” twice.*

The authors would like to thank the reviewer for catching this typo. This has been fixed in the revised manuscript.

Reviewers' Comments:

Reviewer #1:

Remarks to the Author:

I am satisfied with the authors' responses. I find revised manuscript much clearer, better organised in sections and therefore easier to follow-up.

I would have only two more minor comments:

1. In the part you have added in order to better explain "static methods", part 3.1, in sentence "The static techniques used for comparison include the techniques used for comparison include peak-to-peak amplitude of the spike and min-max of its derivative..." part "include the techniques" should be deleted.

2. I think that longer and more descriptive figure captions (as also reviewer 2 suggested) would improve presentation of the work that has been done and highlight the results

Reviewer #2:

Remarks to the Author:

The authors have addressed my comments and I am overall happy with the outcome of this paper. -
Marius Pachitariu

Reviewer #3:

Remarks to the Author:

Reviewers Comments, revised manuscript: On-Implant Online Spike Sorting based on Salient Feature Selection, Nature Communications, NCOMMS-19-15270, by Mohammad Ali Shaeri and Amir M. Sodagar

The authors have improved the manuscript and address many of the problems of clarity. At least I now understand what they did (and more importantly did not do) but in my opinion this contribution alone does not rise to the significance required for Nature Communications.

To summarize, the authors have developed a method for spike classification (assuming a well behaved model generated offline), extracting salient features from that model, and using those features in a proposed Si electronic device to classify spikes more accurately with less circuit footprint and power consumption than previously published strategies. Their strategy is limited to data from single isolated channels and is applied to data from a Blackrock 100 channel array in rhesus macaque. That data is not referenced.

The major deficiency is that there is no complete system description for a Brain Machine Interface that would use this strategy. That complete lack of context undermines the potential impact. There are still many ambiguities. The data origins are still obscure. Are they publicly available? The authors use k-means clustering to create the source of salient features. K-means is among the most input parameter dependent clustering methods, largely abandoned by the community. If we call those sorted units the basis set for salient feature extraction, how many units are found by PCA based or template match spike sorting. How does that outcome impact the salient feature strategy? More importantly, the front end requirements of the spike classifier are completely ignored. How are the spike waveforms to be delivered to the proposed classifier chip? As I stated in my original review, the best commercially available system is 100 times larger and the best published complete system is 40 times larger. The

authors justly argue that these contain some elements not needed for a remote in place front end for their classifier. They also point to there are published designs (have they been fabricated and tested?) for small footprint amplifier designs. However, to have any significance, the authors must point to the best published front end for their classifier, including amplification and filtering, multiplexing, analog to digital conversion, spike detection and spike packet extraction, and then discuss how their design impacts system performance rather than just the classification step.

Finally, I find the current depth of the input data unrealistic. Figure 7 shows that a single salient feature captures 86% (CA) or 65% (CLI) across 512 channels of real data, and one added feature has nearly no effect. I doubt this is possible with any real dataset for which the noise and data variability are accurately represented. Rather than sample some data set, why not acquire access to 512 channels of real Blackrock array data, perform the calculation and then classify those spikes with a well documented spike detection system that could plausibly operate in real time. Classification of an extracted waveforms is so small a part of the online problem, that an advance of that step in the absence of the remainder of the system lacks significance.

I have comments in detail. These are not relevant to a review for which I feel strongly that Nature Communications is not an appropriate venue. An IEEE publication relevant to the actual design or perhaps a neuroengineering journal would be far more appropriate. However, unless the variation of real data is accurately and clearly documented, this report is incomplete. Finally, the title is deceptive. There is no Online or On-implant spike sorting reported. What is reported is a potentially valuable spike classifier run in simulation.

Dear Senior Editor,

The authors would like to sincerely thank you and the reviewers for the time that has been spent on the review of revision-1.

- a) What follows is the reviewers' comments in black, and the authors' responses in blue under each comment.
- b) The revision-2 manuscript being submitted has been prepared with the following color codes (referenced to the revision-1 version):
 - **black**: remains as it is
 - **red**: added/modified according to the reviewers' comments
 - **gray**: deleted (if moved to a different section of the manuscript, it appears in the new location in red.)

Best regards,
M.A. Shaeri and A.M. Sodagar

Reviewers' comments:

Reviewer #1 (Remarks to the Author):

I am satisfied with the authors' responses. I find revised manuscript much clearer, better organised in sections and therefore easier to follow-up.

I would have only two more minor comments:

1. In the part you have added in order to better explain "static methods", part 3.1, in sentence "The static techniques used for comparison include the techniques used for comparison include peak-to-peak amplitude of the spike and min-max of its derivative..." part "include the techniques" should be deleted.

The authors express their gratitude to the reviewer for catching this. This is corrected in the revised manuscript.

2. I think that longer and more descriptive figure captions (as also reviewer 2 suggested) would improve presentation of the work that has been done and highlight the results

Short captions are now longer in the revised manuscript according to this useful comment.

Reviewer #2 (Remarks to the Author):

The authors have addressed my comments and I am overall happy with the outcome of this paper. -
Marius Pachitariu

Reviewer #3 (Remarks to the Author):

The authors have broken the reviewer's comments into 17 separate numbered items in order to clearly provide their answers to each comment, and more easily refer to each individual issue the reviewer has raised.

The authors have improved the manuscript and address many of the problems of clarity. At least I now understand what they did (and more importantly did not do) but in my opinion this contribution alone does not rise to the significance required for Nature Communications. To summarize, the authors have developed a method for spike classification (assuming a well behaved model generated offline), extracting salient features from that model, and using those features in a proposed Si electronic device to classify spikes more accurately with less circuit footprint and power consumption than previously published strategies.

1. Their strategy is limited to data from single isolated channels and is applied to data from a Blackrock 100 channel array in rhesus macaque.

With due respect, the authors disagree with the reviewer on his/her claim that the proposed strategy "is limited to data from single isolated channels". We would like to draw the reviewer's attention to the fact that, basically, our strategy receives spikes on a given channel and performs spike sorting with no pre-assumption or input from other channels. Therefore, it is impossible that isolation or correlation of the channels can affect the proposed strategy in any way. Our proposed idea works for isolated channels as well as for non-isolated ones.

2. That data is not referenced.

This is simply not true. In both the initial submission and revision-1 of the manuscript it is clearly stated that the data used in this work is taken from [25].

3. The major deficiency is that there is no complete system description for a Brain Machine Interface that would use this strategy. That complete lack of context undermines the potential impact.

- The authors would like to refer the reviewer to the following for the context/system description that already exists in revision-1:
 - Page 2-paragraph 1 provides a system-level justification for the necessity of spike sorting in high-density brain implants.
 - On page 2 and 3, Section 1 is all about system description (text + Fig. 1)!

- On pages 3 and 4 (Section 2), a functional block diagram has been provided for the system, describing the spike sorting block on the implant module as well as the blocks supporting the training phase of the proposed approach on the external module.

Moreover, the authors have now added 6.5 additional lines to the caption of Figure 1 (in revision-2) explaining the context of the work at the system level.

- The authors would like to draw the reviewer's attention to the fact that the application of the proposed idea is not restricted to brain-machine interfacing. The reviewer is referred to [Mohan'19, Lemke'19, Kells'19], [Laboy-Juárez'19], [Wood'19], [Stringer'19, Shahidi'19], and [Miyamoto'19], in which spike sorting plays a key role in sensorimotor, somatosensory, auditory, vision, and epilepsy studies, respectively.
- ❖ Mohan, H. et al. Functional architecture and encoding of tactile sensorimotor behavior in rat posterior parietal cortex. *Journal of Neuroscience* 39, 7332-7343 (2019).
- ❖ Lemke, S.M. et al. Emergent modular neural control drives coordinated motor actions. *Nature neuroscience* 22, 1122-1131 (2019).
- ❖ Kells, P.A. et al. Strong neuron-to-body coupling implies weak neuron-to-neuron coupling in motor cortex. *Nature communications* 10, 1575 (2019).
- ❖ Laboy-Juárez, K.J. et al. Elementary motion sequence detectors in whisker somatosensory cortex. *Nature neuroscience* 22, 1438-1449 (2019).
- ❖ Wood, K.C., Town, S.M. and Bizley, J.K. Neurons in primary auditory cortex represent sound source location in a cue-invariant manner. *Nature communications* 10, 1-15 (2019).
- ❖ Stringer, C., et al. High-dimensional geometry of population responses in visual cortex. *Nature* 571, 361-365 (2019).
- ❖ Shahidi, N. et al. 2019. High-order coordination of cortical spiking activity modulates perceptual accuracy. *Nature neuroscience* 22, 1148–1158 (2019).
- ❖ Miyamoto, H. et al. Impaired cortico-striatal excitatory transmission triggers epilepsy. *Nature communications* 10,1917 (2019).

4. There are still many ambiguities. The data origins are still obscure. Are they publicly available?

- Here the reviewer questions the origin of the data again with a different wording. Please refer to the authors' response under Comment-2.
- As for what the CRCNS bank is and who is behind it, we would like to mention an excerpt from the intro of the CRCNS bank (<https://crcns.org/about>) provided below:

“The Collaborative Research in Computational Neuroscience (CRCNS) is *a joint program of NSF and NIH* that, since 2002, has supported integration of theoretical and experimental neuroscience through collaborative research projects typically involving

two to five senior investigators.”

- The specific data set used in our work (<https://crcns.org/data-sets/vc/pvc-1>) has been contributed by Dario Ringach lab, UCLA.
- As for the availability of the data to the public, the answer is “yes”. The reviewer is referred to [25] in the list of references in both the initial submission and revision-1 of the manuscript, in which the hyperlink: <http://crcns.org/data-sets/vc/pvc-1> takes you to the publicly-available data bank.

5. The authors use k-means clustering to create the source of salient features. K-means is among the most input parameter dependent clustering methods, largely abandoned by the community.

With all due respect, the authors disagree with the reviewer, and do not consider “abandoned” a technique the use of which is being repeatedly and recently reported in high-quality journals. As a few examples, we would like to enlist the following:

- ❖ Valencia, D. & Amirhossein, A. “An efficient hardware architecture for template matching-based spike sorting,” *IEEE Transactions on Biomedical Circuits and Systems* **13**, 481-492 (2019).
- ❖ Do, A. T. et al, “An Area-Efficient 128-Channel Spike Sorting Processor for Real-Time Neural Recording With $0.175 \mu\text{W}/\text{Channel}$ in 65-nm CMOS,” *IEEE Transactions on Very Large Scale Integration (VLSI) Systems* **27**, 126 – 137 (2019).
- ❖ Caro-Martín, C. R. et al. “Spike sorting based on shape, phase, and distribution features, and K-TOPS clustering with validity and error indices,” *Scientific Reports* **8**, 1-28 (2018).
- ❖ Petrantonakis, P. C. & Poirazi, P. “A novel and simple spike sorting implementation,” *IEEE Transactions on Neural Systems and Rehabilitation Engineering* **25**, 323–333 (2017).
- ❖ Pachitariu, M. et al. “Fast and accurate spike sorting of high-channel count probes with KiloSort,” In *Advances in neural information processing systems*, 4448-4456 (2016).

6. If we call those sorted units the basis set for salient feature extraction, how many units are found by PCA based or template match spike sorting. How does that outcome impact the salient feature strategy?

The authors tried to guess all the possible questions that the reviewer might have meant by the sentences in this comment, but couldn’t come up with a meaningful one.

- I. Here is our best guess:

The reviewer is interested to compare the number of units detected in the offline training phase of our work with that in a hypothetical design in which offline spike sorting is done based on PCA or template matching; and then see the impact of that on the success of the proposed feature selection technique.

If this is the case, as we know, PCA is a feature extraction method and is unable to determine the number of units! We guess that the reviewer assumes the PCA to be followed by a clustering block. In this case, there are a wide variety of classifiers that can be used for this purpose, and the “number of units” would strongly depend on the choice of the clustering method used. We would like to remind the respectful reviewer that the focus of our manuscript is the salient feature selection strategy, not the offline spike sorting technique used in this work.

- II. Though not expressed in his/her comment, our guess is that the reviewer has had the intention to question the impact of the offline spike sorting technique on the success of the salient feature technique proposed in this work.

In the offline training phase, the most salient features are selected in order to discriminate between the units (spike classes) that have been identified in the offline spike clustering task. **It is of crucial importance to note that**, independently from the accuracy of the offline clustering, success of the proposed approach is determined by how well it categorizes the incoming “un-tagged” spikes in the online spike sorting phase under the classes already suggested by the offline spike clustering. Exact number of the units found by the offline clustering, therefore, does not contribute to the performance of the proposed salient feature strategy at all.

- III. Our main claim in this work is the salient feature technique we have proposed. To assess the merit of the proposed approach, its success in helping achieve more accurate classification results should be compared with that of other feature selection/extraction techniques. As presented in the Results section (Figures 6 &7), performance of the proposed salient feature strategy is compared with 8 other approaches. For a fair comparison, (a) all the approaches are evaluated using the same set of input data (provided by the same offline clustering), and (b) all the feature extraction/selection approaches are followed by the same classifier (Gaussian Naive Bayes classifier).

7. More importantly, the front end requirements of the spike classifier are completely ignored. How are the spike waveforms to be delivered to the proposed classifier chip?

Basically, neural signal processing techniques are usually proposed with the assumption that a typical neural recording front-end provides the signals in the digital domain, and spike sorting methods are not an exception. As it is usually the case, the spike sorting block receives the detected and extracted spikes at the input in the form of consecutive digital samples (over the course of the spikes), each of which being a parallel digital word. The proposed spike sorting approach is no different from other spike sorting methods in terms of front-end requirements. These requirements include a sampling rate of 20-30kSample/Sec., 8-10 bits of resolution, and bandpass filtering bandwidth of 6-8kHz.

The reason the above details were not explained in the paper was that they were too obvious pre-assumptions in spike sorting to the audience of such a highly prestigious journal. Since the two other reviewers seem to be on the same page with the authors, we ask the Editorial Team to kindly advise whether to insert the above explanation in the paper.

8. As I stated in my original review, the best commercially available system is 100 times larger

and the best published complete system is 40 times larger. The authors justly argue that these contain some elements not needed for a remote in place front end for their classifier.

The authors are glad to see that the reviewer agrees with their argument.

9. They also point to there are published designs (have they been fabricated and tested?) for small footprint amplifier designs.

Yes, they are fabricated and tested.

10. However, to have any significance, the authors must point to the best published front end for their classifier, including amplification and filtering, multiplexing, analog to digital conversion, spike detection and spike packet extraction, and then discuss how their design impacts system performance rather than just the classification step.

As for the significance of our proposed idea, as clearly mentioned in the abstract of the manuscript, superiority of the proposed work is two-fold: *“Compared with other similar works, this method shows reduction in classification error by a factor of ~2 (as the improvement in signal processing performance) and also reduction in the required memory space by a factor of ~5 (as the reduction in hardware implementation cost).”*

(a) The considerable reduction in classification error is no doubtedly a signal-level significance, which is interestingly achieved at no additional hardware cost compared to the best work so far published in the context of on-implant spike sorting (ref. [9] of the manuscript). This, in and of itself, is quite an achievement in the context of hardware-efficient embedded signal processing on high-density brain implants. However, when realized in hardware, the proposed technique provides significant advantage in silicon area consumption even when it sits by other system blocks, as explained below.

(b) To provide a rough area estimation at the system level, we should note that the most area-consuming blocks in the signal path are the amplifiers (+bandpass filtering*) and the analog-to-digital converters (ADCs). The recording front-end chip recently published by Cauwenberghs’ group (@ U. California at San Diego) in IEEE JSSC’18 (fabricated and and experimentally tested in-vivo on marmoset monkeys) [Kim’18], consists of neural amplifiers taking per-channel area of ~0.004mm², and advanced oversampling ADCs each taking ~0.02mm². Assuming that each ADC is shared among 16 channels, a 512-channel recording front-end would roughly take a silicon area of 2.688mm² (=0.004×512+0.02×(512÷16)). Spike detection and extraction tasks are both realized in the digital domain. Both tasks together can be implemented using rather simple digital hardware. Using the spike detection and extraction technique the authors already proposed in ref. [2], 512 spike detector and extractor blocks take a total area of 0.23mm²@130nm. The multiplexer is as simple as one transistor per channel and an address decoder, which takes an absolutely negligible area. Now, let us assume two designs: (A) a 512-channel front-end providing signals to a 512-channel version of the on-implant online spike sorter reported in ref. [9] (as the best on-implant spike sorter so far reported in the literature), and (B) the same front-end with our design as the spike sorter (again for 512 channels).

Rough estimation of the total areas:

Design A: $A_{\text{Front-End}} + A_{\text{Spike Sorter in [9]}} = 2.688\text{mm}^2 + (0.023\text{mm}^2/\text{Ch.} \times 512) = 14.464\text{mm}^2$

Design B: $A_{\text{Front-End}} + A_{\text{Spike Det. \& Ext.}} + A_{\text{Our Spike Sorter}} = 2.688\text{mm}^2 + 0.23 + (0.0066\text{mm}^2/\text{Ch.} \times 512) = 6.2972\text{mm}^2$

This means, that even when integrated with front-end circuitry, the proposed on-implant spike sorter still results in a system-level area saving of 56.46%, which is a great significance over the best work so far published in this area.

Putting the above signal-level and circuit-level superiorities together, the proposed idea proves itself as a major step forward in on-implant online spike sorting.

* In the majority of neural amplifiers, the filtering function is embedded in the amplifier circuit.

- ❖ Kim, et al. Sub- μV rms-Noise Sub- μW /Channel ADC-Direct Neural Recording With 200-mV/ms Transient Recovery Through Predictive Digital Autoranging." *IEEE Journal of Solid-State Circuits* 53.11 (2018): 3101-3110.

11. Finally, I find the current depth of the input data unrealistic.

It is unclear to the authors what the reviewer means by “unrealistic depth”.

- As for the term “unrealistic”, the authors would like to refer the reviewer to the Results section, where it explains that the data set used in this work is a realistic recording from a real lab animal in standard experimental conditions, and is available to the public on an internationally-reputable online database. It should be added that the authors have used the entire dataset, with no selection/filtering/categorization in any way whatsoever.
- As for the “depth” of the input data, we have explained that
 - the data set contains around 15,000 spike classes
 - the data has been recorded using a 10x10 Utah array. As we know, with the spacing between the recording sites of this array, recorded activities are highly uncorrelated [Buzsáki’04].
 - signal-to-noise ratio of the input data set varies within the extremely wide range of ~ 0.3 to ~ 22 .
- ❖ Buzsáki, G., Large-scale recording of neuronal ensembles. *Nature neuroscience*, 7(5), 446-451 (2004).

12. Figure 7 shows that a single salient feature captures 86% (CA) or 65% (CLI) across 512 channels of real data, and one added feature has nearly no effect. I doubt this is possible with any real dataset for which the noise and data variability are accurately represented.

The authors would like to explain this same observation in a different way: It is the significance of the proposed salient feature technique that by using only one salient feature, spike classification significantly outperforms other existing classification techniques (Please compare the CA_{CLI} for 1D-SFS with that for other approaches in Figure 8(a)).

It is true that by using more features, between-class discrimination potentially increases, but we should note that the increase in the dimension of the feature space, directly results in

exponentially-increased computational load/complexity. This is why “dimensionality reduction” is usually recommended prior to spike classification. Main aim of dimensionality reduction is to find the minimum number of features using which spike classification can be performed with acceptable accuracy. Significance of the idea we have proposed is in the fact that with only one salient feature (the absolute minimum), spikes are classified with higher accuracy than the best works ever reported: (The work in ref. [9] with 4 features).

Basically, the difference between CA_{CLI} values for 1D-SFS and 2D-SFS is greater than 5% when the GNB classifier is used for online classification (implemented in software). This difference drops to ~2% when online classification is performed using the WD approach. The reason for this performance degradation can be explained as follows: While exhibiting superior classification performance, the GNB technique is of too high hardware complexity to be implemented on a brain implant. The WD approach trades a few percent of classification accuracy for the possibility of power- and area-efficient hardware implementation.

As for the “noise and data variability” in the dataset used, the authors wish that the reviewer had clarified what (s)he meant by “accurate representation (presentation?)”. We believe all the details of the dataset have been clearly presented in the Results section (which is evidently far beyond “noise and data variability”):

Source: [25] (+ hyperlinked online access link)

Electrode array: Utah array

Number of sites: 10×10

Sampling rate: 30 kSample/s

Quantization resolution: 8 bits

Subject: macaque monkeys (*Macaca fascicularis*)

Recording area: primary visual cortex (V1)

Stimulus: natural images

Number of spike classes: ~15,000

Length of extracted spikes: 48 samples (1.6 ms)

Usage of data: both training and testing

Breakdown of data: training=50%, testing=50% (with no overlap)

SNR range: ~0.3 to ~22

SNR average: ~4.5

Number of units per trial: ~1450

Number of units per trial per channel: 2, 3, 4

13. Rather than sample some data set, why not acquire access to 512 channels of real Blackrock array data, perform the calculation and then classify those spikes with a well documented spike detection system that could plausibly operate in real-time.

To the best of the authors’ knowledge, the highest density available through Blackrock Microsystems is 128 channels (please see: <https://www.blackrockmicro.com/electrode-main/>). If the reviewer knows of a 512-channel Blackrock array, the authors would be pleased to receive more information.

As for the dataset used for the validation of the proposed work, the authors are wondering how the reviewer has come to the conclusion that we have “sampled” a data set. On the contrary, we have randomly synthesized 512 channels of inputs in which the entire recording by a 100-

channel Utah array was used without any sampling/selection. This is indeed the common regular practice in the synthesis of input signals to a spike sorter, for instance in ref. [17], [Quiroga'04], and [Franke'15]. There are, however, other works that use both natural and synthetic input data, examples of which are [9], [16], as well as [Yger'18], [Chung'17], [Rossant'16], and [Friedman'15]. The authors are also aware of some spike sorting works that use only natural recorded signals (not synthetic) to test spike sorting algorithms; the issue is that this way, spike classification accuracy can not be extensively studied against within-class variability unless additional noise is intentionally added to the signal.

- ❖ Quiroga, R. Quian, Zoltan Nadasdy, and Yoram Ben-Shaul. "Unsupervised spike detection and sorting with wavelets and superparamagnetic clustering." *Neural computation* 16(8) (2004): 1661-1687.
 - ❖ Franke, Felix, et al. "Bayes optimal template matching for spike sorting—combining fisher discriminant analysis with optimal filtering." *Journal of computational neuroscience* 38.3 (2015): 439-459.
 - ❖ Yger, P. et al. A spike sorting toolbox for up to thousands of electrodes validated with ground truth recordings in vitro and in vivo. *Elife* 7, e34518 (2018).
 - ❖ Chung, J. E. A fully automated approach to spike sorting. *Neuron*, 95(6), 1381-1394 (2017).
 - ❖ Rossant, C. et al. Spike sorting for large, dense electrode arrays. *Nature neuroscience*, 19(4), 634 (2016).
 - ❖ Friedman, A. et al. A multistage mathematical approach to automated clustering of high-dimensional noisy data. *Proceedings of the National Academy of Sciences* 112(14), 4477-4482 (2015).
14. Classification of an extracted waveforms is so small a part of the online problem, that an advance of that step in the absence of the remainder of the system lacks significance. I have comments in detail.

To the authors, the term “the online problem” in this comment refers to the situation where the recording system is not fast enough to pull off the streaming of the recorded signals in the real-time. With all due respect, the authors strongly believe that the major bottleneck in the development of high-density brain implants will be the handling of the huge amount of neuronal data being recorded. Designers of brain implants always face a strict limitation in the bandwidth of wireless data communication imposed by frequency allocation regulations, as well as serious restriction in the power budget that can be allocated for wireless data transmission. Therefore, a great challenge is to reduce/compress the recorded data while preserving the main content of the recordings. The respectful reviewer is referred to the wide spectrum of signal processing solutions, including spike detection, spike extraction, and data compression (both temporal and spatial) published in the recent years. Thus far, from the standpoint of the data compression it provides, spike sorting is perhaps the most effective way of reporting spike occurrences as well as their wave shapes. As such, contrary to the reviewer’s statement, the authors do believe that classification of extracted spike waveforms is a key solution to the online problem. It is indeed an effective solution to the realization of high-density neural recording brain implants capable of streaming the recorded information to the outside world in the real-time.

As for the significance of the proposed spike sorting approach from a system-level perspective,

we have already provided detailed explanation under comment #10.

15. These are not relevant to a review for which I feel strongly that Nature Communications is not an appropriate venue. An IEEE publication relevant to the actual design or perhaps a neuroengineering journal would be far more appropriate.

The authors respect the reviewer's feelings, but review of scientific articles is a logical process that needs to be done based on convincing rationale/reasoning.

Seeing the above expression by the reviewer, we tried to guess a logical reason for the formation of such a feeling, to no avail. The reviewer could possibly question the appropriateness of our paper by asking the following questions (under which we are providing our answers):

- I. ***Is the manuscript out of the scope of the journal?*** Looking at the Aims and Scope of the journal [<https://www.nature.com/ncomms/about/aims>], the authors strongly believe that this work is quite in line with the scope of Nature Communications, where it identifies itself as a multidisciplinary journal. Our work proposes a novel idea in signal processing and pattern recognition (feature selection and spike sorting/classification), heavily employs area-efficient VLSI hardware implementation, and at the system level provides a solution in the context of high-density brain implants with application in leading-edge research in neuroscience.
- II. ***Are the achievements of the paper insignificant?*** From the standpoint of the significance of the manuscript, the authors have extensively highlighted (both in the manuscript and in their response to comments #10 and #14) the achievements of this work in spike sorting performance, in hardware implementation, and also at the system level.
- III. ***Is the manuscript not worth being published in Nature Communications because it is on a subsystem, not a complete system?*** The Nature family (as well as the Cell Press family) does not have such a general policy/criterion for the rejection of papers. As examples, a brief list of recent papers introducing spike sorting techniques and microelectrode arrays are provided below:
 - ❖ Mohanty, A. et al. Reconfigurable nanophotonic silicon probes for sub-millisecond deep-brain optical stimulation. *Nature Biomedical Engineering* (2020): 1-9.
 - ❖ Shin, H. et al. Multifunctional multi-shank neural probe for investigating and modulating long-range neural circuits in vivo. *Nature communications* **10**(1) (2019): 1-11.
 - ❖ Tsai, D. et al. A very large-scale microelectrode array for cellular-resolution electrophysiology. *Nature communications* **8**(1) (2017): 1-11.
 - ❖ Chung, J. E. A fully automated approach to spike sorting. *Neuron*, **95**(6), 1381-1394 (2017).

- ❖ Rossant, C. et al. Spike sorting for large, dense electrode arrays. *Nature neuroscience*, **19**(4), 634 (2016).
- ❖ Park, D.W. et al. Graphene-based carbon-layered electrode array technology for neural imaging and optogenetic applications. *Nature communications* **5** (2014): 5258.

16. However, unless the variation of real data is accurately and clearly documented, this report is incomplete.

In the manuscript we have quantitatively and accurately presented within-class variability (signal-to-noise range and average) and between-class variability (dissimilarity of spike shapes) of the real data we have used. The authors expect the reviewer to provide convincing reasons why (s)he repeatedly questions the accuracy and clarity of the dataset they have used.

17. Finally, the title is deceptive. There is no Online or On-implant spike sorting reported. What is reported is a potentially valuable spike classifier run in simulation.

The authors have been extremely careful in their choice of words in all parts of the paper, and that includes the title as well. We draw the reviewer's attention to the difference between the terms "on-implant online spike sorting" and an "on-implant online spike sorter". Contrary to the reviewer's impression, as the first sentence of the abstract clearly says, "this paper proposes a novel framework for real-time online spike sorting on implantable neural recording micro-systems", not an on-implant spike classifier! The former talks about a spike sorting approach, which is fast enough to operate online and of sufficiently small area and low power consumption to be implemented on brain implants. This is while the latter talks about an online spike sorter block that has been implemented on a brain implant. In the former, the focus is on a signal processing approach, while in the latter the focus is on a circuit block that does spike sorting and has been realized on an implant.

In addition to functional design and simulation at the signal level, hardware implementation of the proposed method has been completely designed using standard VLSI CAD tools. Area consumption of a microelectronic chip is determined upon the completion of physical layout design.

Reviewers' Comments:

Reviewer #1:

Remarks to the Author:

Authors addressed my main concerns. Yet, I would strongly suggest to modify the title, since "Online" is misleading. They develop an interesting novel method, while they do not exploit it yet in a real on-line platform, or during any real experiments.

Something mentioning that this is a "Framework for..." and excluding the word "online" could be more appropriate.